# Notch transactivates Rheb to maintain the multipotency of TSC-null cells

Jun-Hung Cho[1], Bhaumik Patel[1], Santosh Bonala[1], Sasikanth Manne[1], Yan Zhou[2], Surya K. Vadrevu[1], Jalpa Patel[1], Marco Peronaci[1], Shanawaz Ghouse[1], Elizabeth P. Henske[3], Fabrice Roegiers[4], Krinio Giannikou[5], David J. Kwiatkowski[5], Hossein Mansouri[6], Maciej M. Markiewski[1], Brandon White[7] & Magdalena Karbowniczek [1]

Differentiation abnormalities are a hallmark of tuberous sclerosis complex (TSC) manifestations; however, the genesis of these abnormalities remains unclear. Here we report on mechanisms controlling the multi-lineage, early neuronal progenitor and neural stem-like cell characteristics of lymphangioleiomyomatosis (LAM) and angiomyolipoma cells. These mechanisms include the activation of a previously unreported Rheb-Notch-Rheb regulatory loop, in which the cyclic binding of Notch1 to the Notch-responsive elements (NREs) on the Rheb promoter is a key event. This binding induces the transactivation of Rheb. The identified NRE2 and NRE3 on the Rheb promoter are important to Notch-dependent promoter activity. Notch cooperates with Rheb to block cell differentiation via similar mechanisms in mouse models of TSC. Cell-specific loss of Tsc1 within nestin-expressing cells in adult mice leads to the formation of kidney cysts, renal intraepithelial neoplasia, and invasive papillary renal carcinoma.

[1] Department of Immunotherapeutics and Biotechnology, School of Pharmacy, Texas Tech University Health Science Center, 1718 Pine Street, Abilene, TX 79601, USA. [2] Bioinformatics and Biostatistics Facility, Fox Chase Cancer Center, 333 Cottman Avenue, Philadelphia, PA 19111, USA. [3] Division of Pulmonary and Critical Care Medicine, Brigham and Women's Hospital and Harvard Medical School, 45 Francis Street, Thorn Building, Room 826, Boston, MA 02115, USA. [4] Fox Chase Cancer Center, 333 Cottman Ave, Philadelphia, PA 19111, USA. [5] Division of Pulmonary and Critical Care Medicine, Brigham and Women's Hospital, Harvard Medical School and Dana Farber Cancer Institute, 20 Shattuck Street, Thorn Building 826C, Boston, MA 02115, USA. [6] Department of Mathematics and Statistics, Texas Tech University, 1108 Memorial Circle, Lubbock, TX 79409, USA. [7] Department of Biological Sciences, San Jose State University, One Washington Square, San Jose, CA 95192, USA. Correspondence and requests for materials should be addressed to B.W. (email: brandon.white@sjsu.edu) or to M.K. (email: magdalena.karbowniczek@ttuhsc.edu)

The heterogeneity of cancers reflects the aberrant cell differentiation[1, 2]. Poor differentiation of tumor cells often indicates aggressive behavior and stem cell-like properties[3]. The differentiation abnormalities are a hallmark of the central nervous system and peripheral lesions of the tuberous sclerosis complex (TSC), which is a genetic disorder resulting from the loss of TSC1/2 function, manifesting in the form of brain tumors with aberrant glioneuronal differentiation, pulmonary lymphangioleiomyomatosis (LAM), and renal angiomyolipomas[4]. The differentiation plasticity of TSC tumor cells is supported by the expression of melanocytic and smooth muscle markers[5] and the common origin of vascular, smooth muscle, and fat components of angiomyolipoma[6]. However, the mechanisms behind this plasticity are unclear. Since melanocytes and some smooth muscle cells derive from the neural crest (NC) and LAM and angiomyolipoma express melanocytic and smooth muscle markers, we postulate that the mechanisms regulating NC differentiation might also operate in LAM and angiomyolipoma. The Notch signaling pathway regulates NC cell differentiation, maintains neural precursors in an undifferentiated state, and impacts cell proliferation and migration during normal development and in cancer[7–16]. The involvement of Notch in TSC pathogenesis has been suggested by studies demonstrating that Rheb activates Notch in angiomyolipoma-derived cells and that TSC proteins regulate the Notch-dependent cell fate decisions during Drosophila sensory organ development[17, 18]. The oscillation in Notch signaling maintains neuronal progenitors in undifferentiated state[19]. Our data imply that angiomyolipoma cells do not achieve terminal differentiation and remain as neural stem-like cells or progenitors; therefore, we explore the possibility of oscillatory Notch1 signaling gene expression as an underlying mechanism blocking angiomyolipoma cell differentiation.

Here we describe a novel Rheb-Notch-Rheb loop and its role in abnormal differentiation of LAM and angiomyolipoma cells that resemble neural stem cells (NSCs) and neuronal progenitors. The elements of this loop include Rheb, which activates Notch1[17, 18], and the previously unreported direct binding of Notch1 to the Rheb promoter. We identified four potential recombination signal binding proteins for immunoglobulin kappa J region (RBPJ) binding sites within the promoter of Rheb. We discovered that binding of Notch1 to the two Notch1-responsive elements (NREs), NRE2 and NRE3, regulates the transcription of Rheb in a cyclic manner and is essential for Notch-dependent expression of Rheb, indicating that Notch1 is a direct and upstream regulator of Rheb, in addition to the tuberin GTPase-activating protein (GAP) domain[20]. The dysregulation of this mechanism leads to the retention of the NSC-like potential of angiomyolipoma cells and TSC tumorigenesis.

## Results

**Neural crest markers in LAM and angiomyolipoma.** Clinical evidence and the expression of melanocytic and smooth muscle markers point to LAM and angiomyolipoma differentiation plasticity along NC lineages[5, 6, 21]. Other cell types in addition to melanocytes and smooth muscle cells, including neurons and glial cells of the peripheral nervous system, originate from the NC[10]. Therefore, we determined whether the LAM and angiomyolipoma differentiation plasticity involves other NC lineages.

Neuron-specific enolase (NSE) and glial fibrillary acidic protein (GFAP) were expressed in TSC-associated and sporadic angiomyolipoma and LAM, but not in normal adjacent tissue (Fig. 1a–c). Although NSE is not exclusively a neuronal marker, it identifies cells of neuronal and neuroendocrine origin. The expression of neuron-specific tubulin (NS-tubulin) within small clusters of angiomyolipoma supports the neuronal or melanocyte

nature of these cells (Fig. 1b and Supplementary Fig. 1A)[22]. In addition to angiomyolipoma, the expression of NS-tubulin was present in papillary micro adenoma from the same patient (Supplementary Fig. 1A, fourth panel). In the normal kidney NS-tubulin staining was detected only in peripheral nerves, as it should be, confirming high specificity of this assay (Supplementary Fig. 1A, first panel). Nestin, a typical NSC marker[23], also detected in various cancer cells of neuronal and non-neuronal origin[24], was expressed in small angiomyolipoma clusters (Fig. 1b), and LAM cells (Fig. 1d). The expression of GFAP, NSE, and nestin in available angiomyolipoma tumors and lack of or very low expression in corresponding normal kidneys was confirmed by western immunoblotting (Fig.1c(i, ii)), Supplementary Fig. 1E-ii and Supplementary Fig. 10). Nestin and the neuronal marker peripherin[25] were co-expressed in LAM and angiomyolipoma, but not in normal adjacent cells (Fig. 1d, Supplementary Fig. 1B and Supplementary Table 1).

The differentiation plasticity of angiomyolipoma is supported by the co-expression of multiple NC markers[26, 27] in angiomyolipoma-derived cell lines[28–30] determined by fluorescence-activated cell sorting (FACS) (Fig. 1e and Supplementary Fig. 1C). For comparison, we used NC-derived melanoma cells (Supplementary Fig. 1D), expecting that NC markers will be expressed in these cells. Tuberin (TSC1 product) and hamartin (TSC2 product) expression was examined by western blotting (WB) in angiomyolipoma-derived cells (Supplementary Fig. 1E). The upregulation of mammalian target of rapamycin (mTOR) signaling in angiomyolipoma cells was confirmed by WB (Supplementary Fig. 1E, F, and H), real-time reverse transcriptase-PCR (RT-PCR) (Supplementary Fig. 1G), and global transcriptomic analysis of angiomyolipoma and angiomyolipoma-derived cells vs. the corresponding normal kidney (Supplementary Fig. 2 and Supplementary Table 2).

Through FACS (Fig. 1e(i) and Supplementary Fig. 1C-D), we have identified subpopulations of cells with profiles similar to either the NSC-like cells or multipotent progenitors. These populations consist of: (1) cells expressing nestin alone, similar to NSC[10, 31]; (2) cells co-expressing nestin and melan-A, or NS-tubulin, indicating melanocyte or neuronal differentiation, respectively; and (3) cells co-expressing nestin, GFAP, and melan-A, reflecting glial or melanocyte differentiation[32] (Fig. 1e (i) and Supplementary Fig. 1C). The expression of nestin, NS-tubulin, and GFAP in angiomyolipoma-derived cells was verified by WB (Fig. 1e(ii)). The NSC-like profile of angiomyolipoma cells was inter-related with the expression of transcription factors maintaining the multipotency of NSC and preventing differentiation, such as SoxE and inhibitor of DNA binding (ID) subfamily[2, 33], SOX10, SOX9, and ID3 (Fig. 1f). Angiomyolipoma cells also have the potential to differentiate along Schwann and melanocytic lineages, since they expressed S100A1 (Schwann), DCT, and c-Kit (melanocytic) markers[34, 35] (Fig. 1f). These data suggest that angiomyolipoma cells with mutated TSC1/2[28–30] and/or hyperactivated Rheb[28–30] (Supplementary Fig. 1E–H) have a similar profile to NSC-like cells and early multipotent progenitors.

To support differentiation plasticity of angiomyolipoma through unbiased approach, we performed a global transcriptomic analysis of sporadic angiomyolipoma, corresponding normal kidney, and two angiomyolipoma-derived cell lines using RNA sequencing (RNA-seq). Three gene lists were intersected, and only common genes which were differentially expressed in angiomyolipoma and angiomyolipoma cells relative to normal kidney (Supplementary Fig. 2) were used for the gene ontology (GO) enrichment analysis. The analysis of the significant genes was performed using GOstats package (Bioconductor)[36]. The enriched biological process were defined by P-value cutoff of <0.001 using hypergeometric test with GOstats package, and only

the enriched categories related to neurogenesis, multipotency, and differentiation are shown. These processes include axon extension, regulation of neuron and cell differentiation, morphogenesis, and nervous system development (Supplementary Fig. 3). Using pathway analysis, we found alterations in expression of genes involved in stem cell pluripotency, melanocyte pigmentation, and renal cell carcinoma (RCC) signaling (Supplementary Table 2). In addition, the bio-function categories enriched for differentially expressed genes in angiomyolipoma tumor and angiomyolipoma cell lines are shown (Supplementary Table 3). In conclusion, these analyses point to differentiation plasticity of

angiomyolipoma and potential of tumor cells to differentiate into NC-derived tissues and cells.

Since activation of mTOR is the driver of TSC tumorigenesis, we assessed the effect of rapamycin (an mTOR inhibitor) on the NSC-like profile of angiomyolipoma cells. Although rapamycin reduced mTORC1 activation because it decreased the number of cells containing phosphorylated ribosomal protein 6 (pS6), which is mTORC1 downstream target (Supplementary Fig. 5A), the number of nestin-positive NSC-like cells did not change (Supplementary Fig. 5B). Instead, rapamycin shifted the quiescent angiomyolipoma NSC-like cells (nestin$^+$Ki67$^-$) from G$_{alert}$

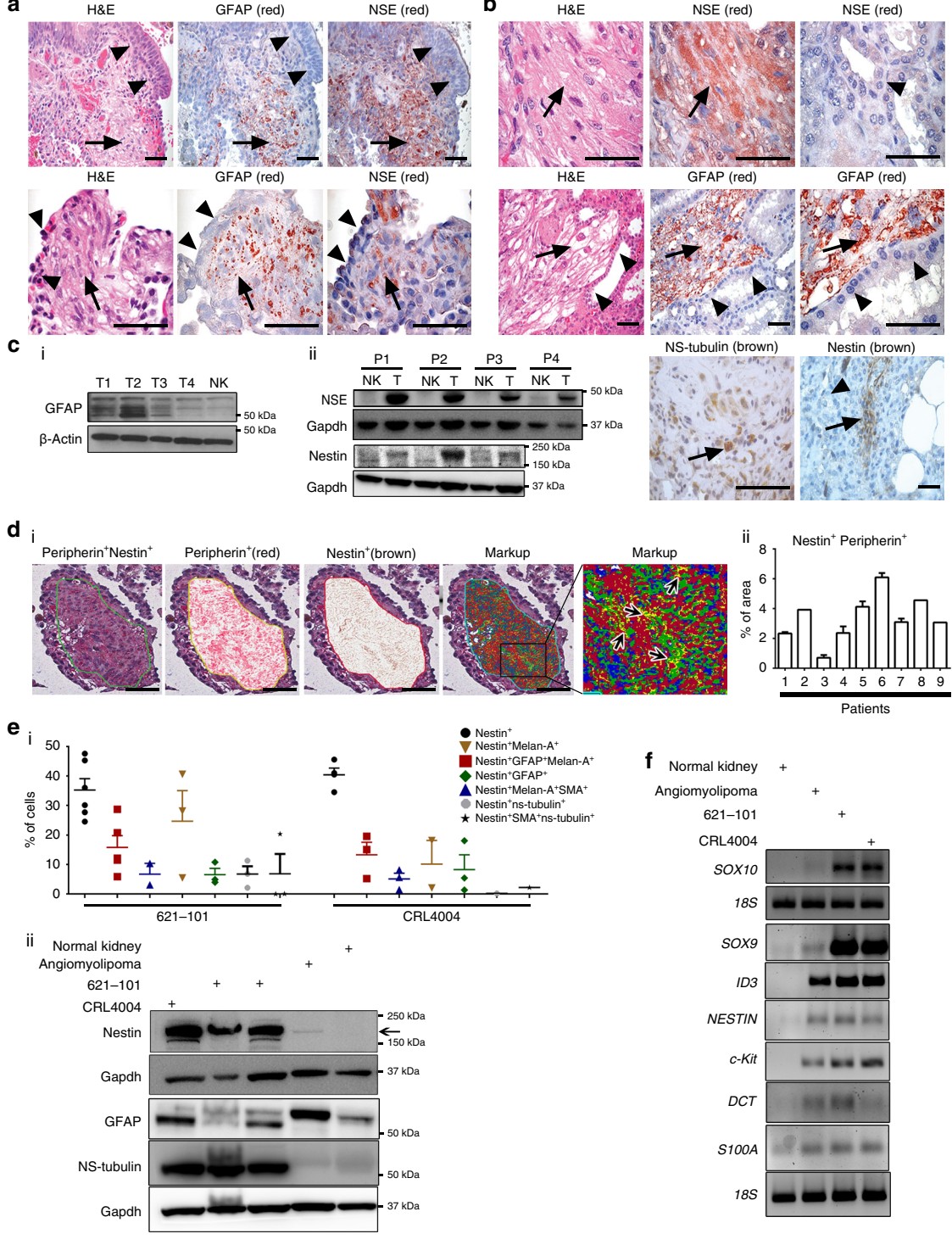

(nestin[+]Ki67[−]pS6[+]) to $G_0$ (nestin[+]Ki67[−]pS6[−]) (Supplementary Fig. 5C), consistent with the role of mTORC1 in 'alerting' quiescent stem cells[37]. These data also suggest that aberrant differentiation of angiomyolipoma is not caused by mTOR upregulation.

**The oscillation of *RHEB* and *HES1* in angiomyolipoma cells.** Oscillatory gene expression regulates cell cycle[38], somitogenesis[39], and neural progenitor differentiation[19]. Given the role of Notch signaling oscillation in maintaining the multipotency of NSCs and progenitors[19] in conjunction with the phenotype of angiomyolipoma cells (Fig. 1 and Supplementary Figs. 1–3), and roles of Rheb and Notch in TSC tumorigenesis[17, 18], we theorized that cycling of Rheb and Notch signaling in NSC-like angiomyolipoma cells prevents their differentiation. We looked at the endogenous *RHEB* and *HES1* messenger RNAs (mRNAs) by quantitative reverse transcriptase-PCR (q(RT)-PCR) in angiomyolipoma-derived cells, synchronized as previously described[40, 41]. *RHEB* mRNA oscillated, with peaks at 30, 120, and 210, or 60, 120, and 240 min, after addition of serum, which initiated oscillation (Fig. 2a(i, iii), Supplementary Fig. 6A, and Supplementary Fig. 8A-i, B-i), with some variability between used cell lines, and reached a maximum at 120, 210, or 240 min (Fig. 2a(i, iii) and Supplementary Fig. 6A, 8A-i, B-i). *HES1* mRNA oscillated with peak at 90 min (Fig. 2a(ii, iv) Supplementary Fig. 6B, and Supplementary Fig. 8A-ii, B-ii). The observed *RHEB and HES1* oscillation was not related to changes in *GAPDH* (Supplementary Fig. 6C and D, black lines). We corroborated qPCR data by WB (Fig. 2b(i-ii) and Supplementary Fig. 10). The oscillation of Hes1 depends on Notch signaling[19]. Therefore, similarity in the oscillation patterns (regardless of differences in the magnitude of oscillation) of Rheb and Hes1 led us to hypothesize that Notch1 also regulates the changes in *RHEB* mRNA expression. We searched for potential RBPJ binding sites within the human Rheb promoter, since these elements bind the Notch1 ternary complex, to regulate the target gene promoters[42]. We identified four potential RBPJ binding sites, with the last site located within the exon 1 (Fig. 2c(i)). We used chromatin immunoprecipitation qPCR (ChIP-qPCR) to examine the binding of Notch1 to the promoter of Rheb and Hes1 (as a positive control) in angiomyolipoma cells in Dulbecco's modified Eagle's medium (DMEM) to identify which sites are functionally important and correspond to true NREs. We also measured the activity of the Rheb promoter using luciferase-reporter gene assays in the presence or absence of the Notch1 ternary complex members: mastermind-like 1 (MAML1) and Notch1 intracellular domain (N1ICD). Notch1 bound mainly to the NRE2 and NRE3 within the Rheb (Fig. 2c(ii, iv)) and Hes1 promoters (Fig. 2c(iii, v)). In addition, Rheb reporter expression was increased by

N1ICD and MAML1 (Fig. 2d(i)), supporting the direct activation of Rheb transcription through NRE2, NRE3, and Notch1.

To determine which promoter elements are essential for the transcriptional activation of Rheb, we altered these potential RBPJ binding sites including NRE2 and NRE3 (one at a time) as previously described[43]. We measured the fold activation of each mutant relative to the activity of the wild-type promoter of Rheb (Fig. 2d(i)). Mutation of the potential RBPJ binding site 1 or 4 did not change the basal and N1ICD-induced promoter activity (Fig. 2d(i)). The mutation in NRE2 or NRE3 significantly suppressed the basal Rheb promoter activity, and the expression of MAML1 and N1ICD was able to raise activity of these mutants only to the level of basal wild-type Rheb promotor (Fig. 2d(i)). The activity of NRE2 and NRE3 double mutant was similar to NRE2 or NRE3 single mutants (Fig. 2d(i)), suggesting that two sites are required for Notch1-dependent transactivation of Rheb and that altering either one is sufficient to eradicate this regulation. These results also suggest that Notch1 regulates approximately 50% of Rheb promoter basal activity. To confirm that Notch1 transactivates Rheb, we overexpressed N1ICD and MAMl1 or downregulated Notch1 using shRNA in 293T, HeLa, CRL4004, 621–101, and H1299 cells. Overexpression of N1ICD and MAML1 increased the expression of endogenous Rheb by 2-fold on average, whereas downregulation of Notch1 reduced Rheb expression by 5-fold (Fig. 2d(ii-iv) and Supplementary Fig. 10). Next, we generated focal deletion of the NRE3 region within the endogenous *RHEB* promoter in CRL4004 angiomyolipoma cells using CRISPR/Cas9. These deletions suppressed expression and activation of Rheb (Supplementary Fig. 4), confirming a direct regulation of Rheb by Notch1.

Our data indicate that Notch1 activates Rheb, and therefore we hypothesized that the oscillation in expression of Rheb (Fig. 2a, b and Supplementary Fig. 10) results from the oscillatory interaction of Notch1 with NRE2 and NRE3. We examined binding of Notch1 to the promoter of Rheb at several time points after cells in DMEM were released from the synchronization conditions[40, 41]. Notch1 did not bind to the potential RBPJ binding site 1 (Fig. 2e(i) and Supplementary Fig. 8c(i)); in contrast, it bound in an oscillating manner to NRE2 and NRE3 (Fig. 2e-ii-iii and Supplementary Fig. 8C-ii-iii). The level of Notch1 binding (Fig. 2e (ii, iii) and Supplementary Fig. 8C-ii-iii) correlated with the increase in *RHEB* mRNA levels (Fig. 2a(i)). These data support that NRE2 and NRE3 regulate Rheb expression, in contrast to the potential RBPJ binding site 1, which does not appear to play a role in the regulation of Rheb.

Next, we determined whether the similar mechanism regulates the oscillation of Hes1. We found that Notch1 binds to the Hes1 promoter in an oscillatory manner (Fig. 2e(iv) and Supplementary Fig. 8C-iv), as it was a case for Rheb promoter. Although

**Fig. 1** Multipotent progenitor and NSC-like profile of LAM and angiomyolipoma. **a** Hematoxylin and eosin (H&E) of the lung sections from sporadic LAM; GFAP or NSE expression in LAM. Arrows indicate LAM and arrowheads normal bronchial mucosa (upper) or normal pleura (lower images). **b** Sporadic angiomyolipomas (upper left and upper middle) and adjacent normal kidney (upper right) or TSC-associated (lower images) angiomyolipomas, stained with H&E or for NSE, GFAP, NS-tubulin, or nestin (arrows indicate angiomyolipoma and arrowheads kidney). **c** (i) GFAP, (ii) NSE and nestin expression in (i) TSC-associated and sporadic (ii) angiomyolipomas (T) from patients (P) vs. normal kidney (NK) by western immunoblotting. **d** (i) Nestin and peripherin expression in LAM (first left) and co-expression analysis by digital pathology; red indicates peripherin (second), brown nestin (third image), and yellow indicates co-expression of both markers (markup, right fifth panel: details within the area marked by the corresponding insert in fourth panel, black arrows). (ii) Quantification for data shown in **d** (i) by Aperio Digital Pathology; the percentage of area with co-expression of peripherin and nestin in LAM from 9 patients. Error bars are defined as means + s.e.m. **e** (i) FACS of cells grown in DMEM medium. Percentages of 621–101 and CRL4004 (angiomyolipomas-derived) cells expressing nestin alone or co-expressing: melan-A, GFAP, SMA, and NS-tubulin; (ii) Nestin, GFAP, and NS-tubulin expression in 621–101 (grown in DMEM or IIA complete medium), CRL4004 angiomyolipoma cells, angiomyolipoma tumor, and corresponding normal kidney by western immunoblotting. Error bars are defined as means + s.e.m. **f** q(RT)-PCR of *SOX10, SOX9, ID3, NESTIN, c-Kit, DCT,* and *S100A1* mRNA levels relative to *18S* in 621–101, CRL4004, angiomyolipoma tumor, and corresponding normal kidney. Data represent means (±s.e.m). Data from nine LAM and eleven angiomyolipomas (**a**–**d**). Scale bar: 50 μm. *$P \leq 0.05$, t-test. Data are representative of three to six (**e** (i)) independent experiments

Hes1 oscillation was shown to be regulated by Notch through use of DAPT (N-[N-(3,5-difluorophenacetyl)-L-alanyl]-S-phenyl-glycine t-butyl ester)[19], we additionally report that *Hes1* oscillation occurs by direct cyclic binding of Notch1 to its promoter.

These results suggest that Notch1 regulates Rheb via binding to NRE2 and NRE3. In conjunction with the Rheb ability to activate

Notch1[17, 18], our data also point to a new regulatory circuit involving Notch and Rheb, which we termed the Rheb-Notch-Rheb loop. Since oscillation of Notch maintains multipotent NSCs, we propose that the Rheb-Notch-Rheb loop, controlled by fluctuations in the binding of Notch1 to Rheb-activating and Hes1-activating NREs, contributes to maintaining multipotent properties of angiomyolipoma cells.

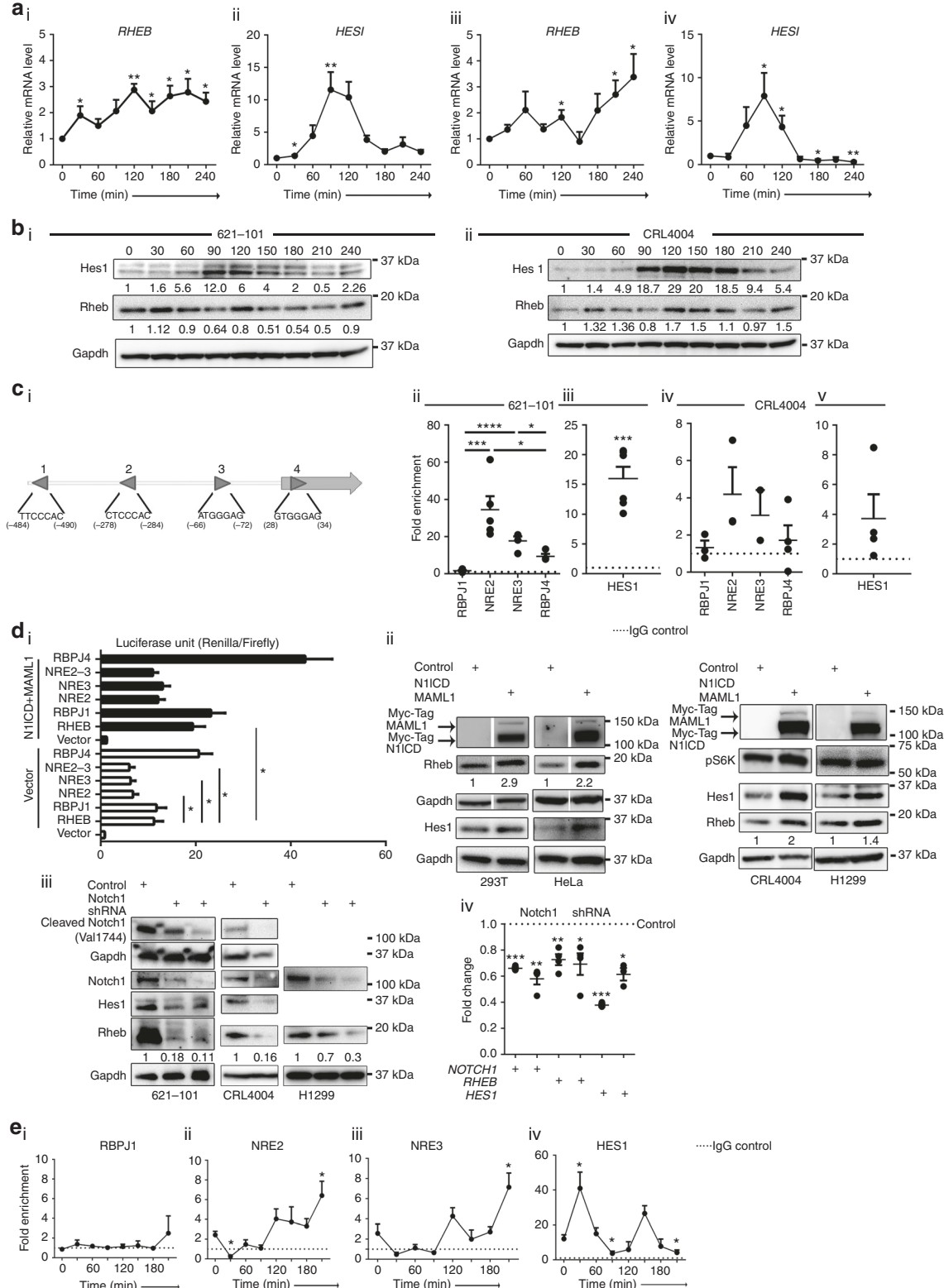

**Neuronal differentiation of angiomyolipoma cells**. Since angiomyolipoma cells express markers of NC lineages, we examined the ability of these cells to differentiate under conditions favoring glial–neuronal differentiation. We used N-medium promoting differentiation of nestin-positive and carotid body-derived cells toward glomus cells[44]. In this medium viabilities of angiomyolipoma cells were 79 and 66% on days 4 and 7, respectively, and they formed sphere-like structures (Fig. 3a, b, d, e), similar to carotid body stem cell neurospheres[44]. The similar structures were formed by NC-derived melanoma cells (Supplementary Fig. 5D) that express the neuronal marker peripherin[45].

The loss of nestin in angiomyolipoma cells during the formation of sphere-like structures (Fig. 3b, c(i), e, f(i)) correlated with the adoption of an early neuronal or melanocyte fate, indicated by increases in percentages of NS-tubulin-positive cells (Fig. 3b, c(ii), e, and f(ii)) and cells co-expressing NS-tubulin and melan-A (Supplementary Fig. 5E), respectively, suggests neuronal traits. The adoption of these fates correlated with an exit from the cell cycle, indicated by the decreased number of Ki-67-positive cells (Fig. 3c (iii), f(iii)), and the induction of neuronal (*Ngn1*) and melanocytic (*DCT* and/or *c-Kit*) differentiation genes (Fig. 3c(iv), f(iv)).

To determine whether hyperactivation of Rheb and Notch prevents neuronal differentiation, we used short hairpin RNA (shRNA) to deplete angiomyolipoma cells of Rheb[46] or Notch1. This depletion accelerated the formation of neural-like spheres (Supplementary Fig. 5F-i, G-i) and differentiation along the neuronal lineage in N-medium, demonstrated by faster acquisition of NS-tubulin (Supplementary Fig. 5F-ii, G-ii). In contrast, overexpression of the constitutively active form of Rheb (Q64L)[47] blocked neuronal differentiation of angiomyolipoma cells in N-medium, as the number of NS-tubulin-positive cells was reduced by 2-fold (Supplementary Fig. 5H). Of note, transient over-expression of Rheb does not allow for prolonged (up to 7 days) follow-up of cell phenotypes due to lack of chromosomal plasmid integration. Therefore, this experiment involved only 24 h of exposure to N-medium, after 24 h of transfection (48 h total). The differentiation along the neuronal or melanocyte lineage was not affected by rapamycin (Supplementary Fig. 5I) suggesting again a distinct role of the Rheb-Notch pathway in this process, independent from mTOR signaling.

**Rheb-Notch-Rheb loop in angiomyolipoma differentiation**. We hypothesized that alterations in *RHEB* and *HES1* expression are required for neuronal differentiation of angiomyolipoma cells. Therefore, we examined expression of *RHEB* and *HES1* mRNA and binding of Notch1 to NREs within the Rheb and Hes1 promoters in N-medium. Relative to DMEM, N-medium repressed the expression of both *RHEB* and *HES1* mRNA and block oscillation (Fig. 4a(i, ii), Supplementary Fig. 6A–B and

Supplementary Fig. 8A–B). This effect was present in synchronized cells regardless of type of stimulation, serum (Fig. 4a, green lines), or chick embryo extract (Fig. 4a, Supplementary Fig. 6A–B, red lines, and Supplementary Fig. 8A–B, red dots). ChIP-qPCR revealed that in conditions favoring neuronal differentiation (N-medium), binding of Notch1 to NRE2 and NRE3 on the Rheb promoter was significantly reduced compared to DMEM (Fig. 4b (i, ii) and Supplementary Fig. 8C). We did not detect significant binding of Notch1 to the potential RBPJ binding site 1 in N-medium (Fig. 4b(ii)). The binding of Notch1 to the Hes1 promoter was also reduced in N-medium (Fig. 4b(iii)). WB and qPCR confirmed that N1ICD, Rheb, and Notch target gene *HES5* were reduced in angiomyolipoma cells in N-medium (Fig. 4c, d, f, g and Supplementary Fig. 10). The suppression of Rheb was also demonstrated by reductions in the percentage of phospho-S6-positive cells (Fig. 4e, h), reduced phospho-S6-K (Fig. 4g and Supplementary Fig. 10), and decreased expression of Rheb (Fig. 4f, g and Supplementary Fig. 10). The neuronal differentiation of angiomyolipoma cells was associated with reduced SOX9 and ID1 and ID3 (Fig. 4i), consistent with the role of Notch in the activation of these genes[48], a role for SOX9 in maintaining multipotent NSCs[49], and Id proteins in preventing neurogenesis in neural tube (NT)[50, 51].

We propose that decreased binding of Notch1 to NREs may lead to suppression of Rheb and Hes1 expression and decreased oscillation, leading to sustained, low level of Rheb and Hes1, followed by neuronal differentiation of angiomyolipoma cells. Given that the suppression of Rheb correlated with the suppression of Notch1 during neuronal differentiation of angiomyolipoma cells, we conclude that inhibition of the Rheb-Notch-Rheb regulatory loop is required for this process. Conversely, ongoing activity of this loop maintains multipotent properties of angiomyolipoma cells in DMEM.

**The loss of *Tsc1* delays the differentiation of NSCs in vivo**. The differentiation abnormalities in murine *Tsc*-null neuroepithelial progenitors were reported[52, 53]. Therefore, we next determined whether the loss of *Tsc1/2* blocks progenitor differentiation during mouse embryogenesis via a mechanism similar to that in angiomyolipoma cells. We used a genetic strategy for cell-specific depletion of *Tsc1* in the mouse embryo NT because expression of *ID1* and *ID3* was suppressed in angiomyolipoma cells in N-medium (Fig. 4i), Id proteins as well as high level of Hes1 prevent neurogenesis in NT[50, 51], and Id3 and Hes1 are regulated by Notch1[54]. We utilized lineage tracing in *ROSA^{mT/mG}* mice (Fig. 5) with a floxed tdTomato (mT)-cassette and a stop codon upstream of the EGFP (mG)-cassette (Fig. 5a, b and Supplementary Fig. 7A–C). This system along with Cre strains allows comparison of the control- and mutant-EGFP⁺ cells. We employed a nestin or

**Fig. 2** Notch1 directly activates the Rheb promoter. **a** q(RT)-PCR of (i, iii) *RHEB* and (ii, iv) *HES1* relative to *GAPDH* in synchronized (i, ii) 621–101 and (iii, iv) CRL4004 angiomyolipoma cells in DMEM. The statistical significance of *RHEB* and *HES1* expression at different time points was determined relative to time 0. **b** Expression of Rheb and Hes1 in synchronized (i) 621–101 and (ii) CRL4004 angiomyolipoma cells in DMEM by western immunoblotting. Numeric values represent densitometry analysis of the expression of Rheb relative to expression of Gapdh. **c** (i) Rheb promoter: triangles indicate potential RBPJ binding sites with a reverse (1, 2) and a forward (3, 4) orientation. (ii–v) Binding of Notch1 to (ii, iv) the potential RBPJ binding site 1, NRE2, NRE3, and the potential RBPJ binding site 4 of *RHEB* or (iii, v) *HES1* promoter in non-synchronized (ii, iii) 621–101 or (iv, v) CRL4004 cells grown in DMEM by ChIP-qPCR; binding is shown as fold enrichment over the IgG (dotted line). **d** (i) Wild-type Rheb-luciferase (RHEB), RHEB-RBPJ1- (RBPJ1), RHEB-NRE2- (NRE2), RHEB-NRE3- (NRE3), or RHEB- RBPJ4- (RBPJ4) mutant-luciferase or control-luciferase (control) promoters activity in HeLa cells. (ii) Expression of N1ICD, MAML1, Hes1, Rheb, and phospho-S6K in 293T, HeLa, CRL4004, and H1299 cells. (iii) Expression of cleaved Notch1 (Val1744), Notch1, Hes1, Rheb in 621–101, CRL4004, and H1299 cells depleted of Notch1 using shRNA by western blotting or (iv) q(RT)-PCR. Numeric values represent densitometry analysis of the expression of Rheb relative to expression of Gapdh. (iv) *NOTCH1*, *RHEB*, and *HES1* relative to *GAPDH* in 621–101 cells depleted of Notch1 using shRNA. **e** Binding of Notch1 to the (i) potential RBPJ1 site, (ii) NRE2, and (iii) NRE3 within Rheb or (iv) Hes1 promoter in synchronized 621–101 cells in DMEM by ChIP-qPCR as in (**c** (ii, iii)). Data represent means ± s.e.m. Error bars are defined as means + s.e.m. *$P \leq 0.05$, **$P \leq 0.01$, ***$P \leq 0.001$, *t*-test. Data are representative of three (**a** (iii, iv), **b**, **d** (i–iv)), four (**e**), and six (**a** (i, ii), **c** (ii, iii)) independent experiments

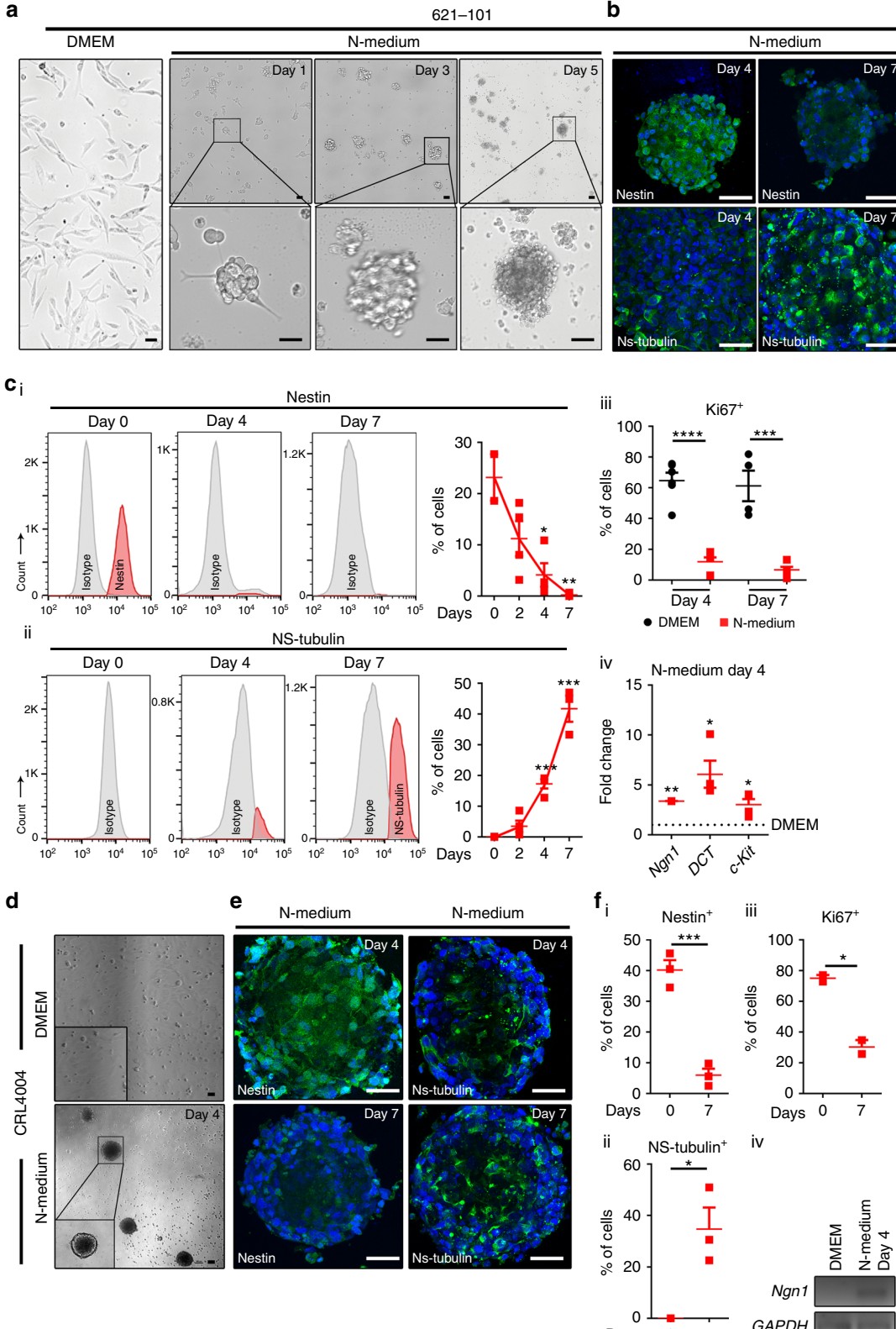

**Fig. 3** Neuronal differentiation of angiomyolipoma cells. **a**, **d** Brightfield images of **a** 621–101 and **d** CRL4004 cells in DMEM or N-medium. **b**, **e** Nestin and NS-tubulin in **b** 621–101 and **e** CRL4004 sphere-like structures in N-medium at day 4 and day 7 by immunofluorescence. **c**, **f** FACS of **c** 621–101 and **f** CRL4004 cells in N-medium. **c**, **f** Representative histograms and graphs showing percentages of cells in N-medium expressing (**c** (i), **f** (i)) nestin only and (**c** (ii), **f** (ii)) NS-tubulin only. **c** (iii), **f** (iii) Ki-67 in (**c** (iii)) 621–101 and (**f** (iii)) CRL4004 cells in N-medium vs. DMEM at days 4 and 7. **c** (iv), **f** (iv) q(RT)-PCR of *Ngn1*, *DCT*, and *c-Kit* mRNA relative to *GAPDH* in **c** (iv) 621–101 or **f** (iv) CRL4004 cells in N-medium vs. DMEM at day 4. Data represent means ± s.e.m. Error bars are defined as means + s.e.m. *$P \leq 0.05$, **$P \leq 0.01$, ***$P \leq 0.001$, ****$P \leq 0.0001$ by *t*-test. Data are representative of two (**b**, **e**), three, $n = 3$ (**f** (i–iv), **c**(iv)), four (**d**); $n = 4$ (**c** (i–iii)), and seven (**a**) independent experiments

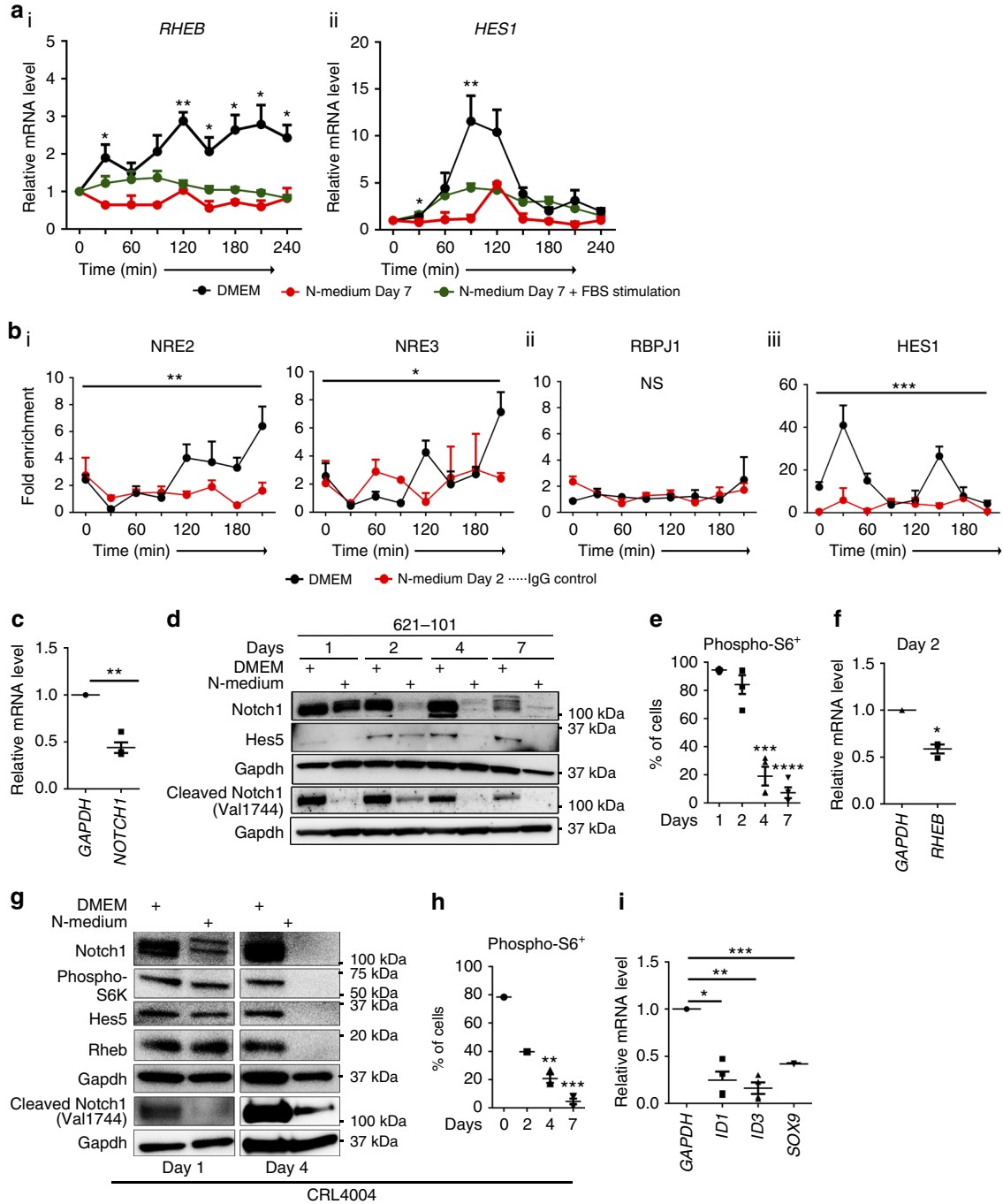

**Fig. 4** Suppression of Rheb and Hes1 oscillation associates with neuronal differentiation of angiomyolipoma cells. **a** q(RT)-PCR of (i) *RHEB* and (ii) *HES1* relative to *GAPDH* in synchronized 621–101 cells in N-medium (released with chicken embryo extract (red lines) or serum (green lines)) vs. DMEM (released with serum (black lines)) (data from DMEM are also used in Fig. 2a (i–ii)). The statistical significance of *RHEB* and *HES1* expression in DMEM vs. N-medium was determined between corresponding time points. **b** Binding of Notch1 to the (i) NRE2, NRE3, and (ii) potential RBPJ1 site within the Rheb or (iii) Hes1 promoter, in synchronized 621–101 cells in N-medium vs. DMEM at day 2 by ChIP-qPCR (data from DMEM are also used in Fig. 2e (i–iv)). Asterisks represent the statistical significance of mean of time points in DMEM vs. N-medium, two-way ANOVA. **c** q(RT)-PCR of *NOTCH1* in 621–101 cells in N-medium at day 2, relative to DMEM (control) after normalization to *GAPDH*. **d** Western blot of Notch1, Hes5, and cleaved Notch1 (Val1744) in 621–101 cells in N-medium vs. DMEM. **e** Percentage of 621–101 cells expressing phospho-S6 in N-medium by FACS. **f** q(RT)-PCR of *RHEB* in 621–101 cells in N-medium at day 2, relative to DMEM (control) after normalization to *GAPDH*. **g** Western blot of Notch1, phospho-S6K, Hes5, Rheb, and cleaved Notch1 (Val1744) in CRL4004 cells in N-medium vs. DMEM. **h** Percentage of CRL4004 cells expressing phospho-S6 grown in N-medium by FACS. **i** q(RT)-PCR of *ID1*, *ID3*, and *SOX9* in 621–101 cells in N-medium at day 2, relative to DMEM (control) after normalization to *GAPDH*. Data represent means ± s.e.m. Error bars are defined as means + s.e.m. *$P \leq 0.05$, **$P \leq 0.01$, ***$P \leq 0.001$, ****$P \leq 0.0001$, t-test (**a**, **c**, **e**, **f**, **h**, **i**), two-way ANOVA (**b**). Data are representative of two, $n = 2$ (**i** for *SOX9*); three (**g**); three, $n = 2$ (**a**, **b** (iii)); three, $n = 3$ (**h**); three, $n = 4$ (**b** (i, ii)); four (**d**), four, $n = 3$ (**f**); four, $n = 4$ (**c**, **e**, **i** for *ID1*, *ID3*)

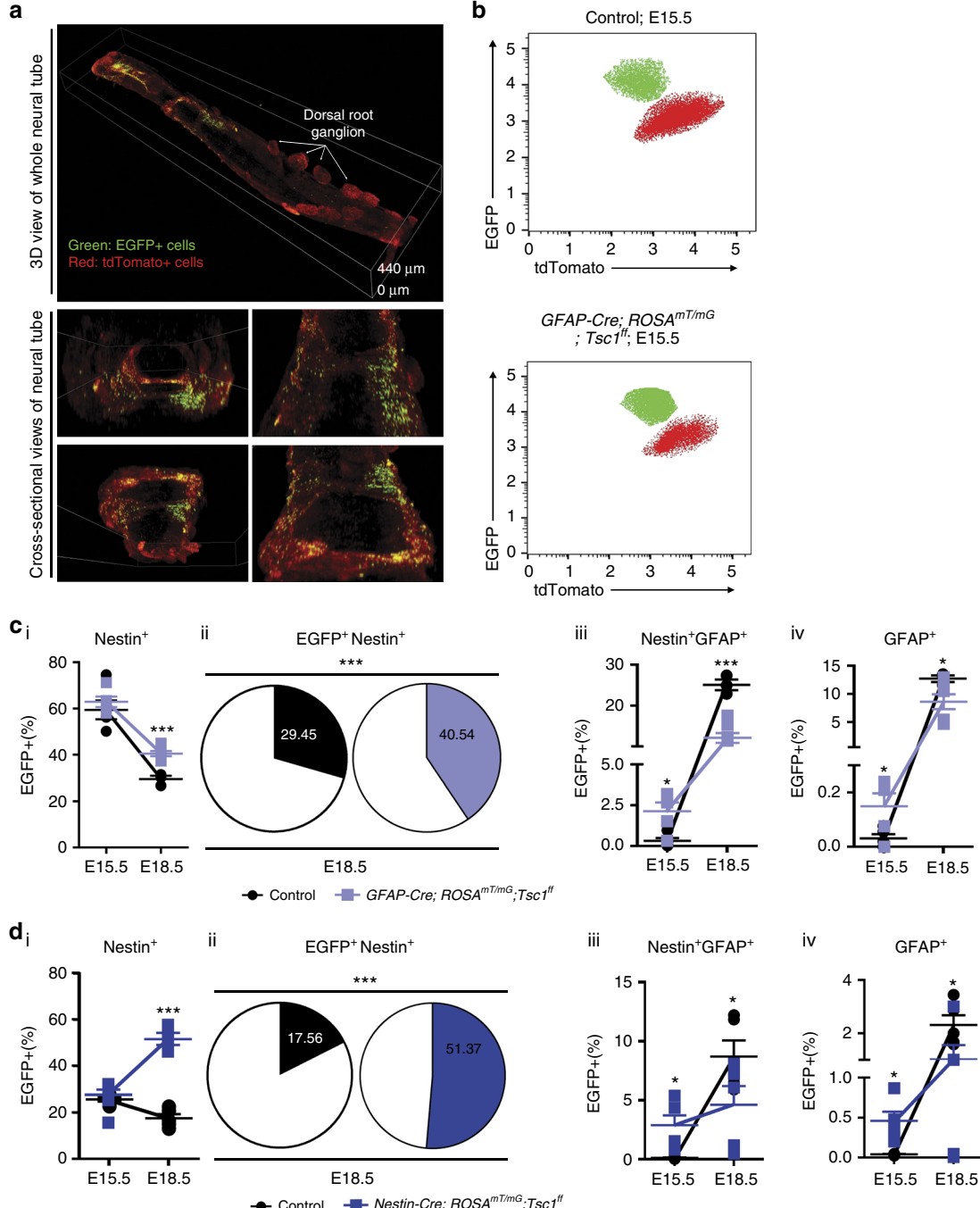

**Fig. 5** Loss of *Tsc1* delays the differentiation of NSCs and RGCs. **a** Confocal microscopy of EGFP-positive (EGFP+) and tdTomato-positive (tdTomato+) NT cells in a *ROSA^mT/mG* mouse NT. Maximum intensity projections from Z-stacks is shown. **b–d** FACS of EGFP+ NT cells in **b**, **c** *GFAP-Cre;ROSA^mT/mG* (control) vs. *GFAP-Cre;ROSA^mT/mG;Tsc1^ff* embryos or **d** *Nestin-Cre;ROSA^mT/mG* (control) vs. *Nestin-Cre;ROSA^mT/mG;Tsc1^ff* embryos. **b** Representative FACS plots of EGFP+ or tdTomato+ NT cells. **c**, **d** *GFAP-Cre;ROSA^mT/mG* (control-black) vs. *GFAP-Cre;ROSA^mT/mG;Tsc1^ff* embryos (blue) (**c**) or *Nestin-Cre;ROSA^mT/mG* (control-black) vs. *Nestin-Cre;ROSA^mT/mG;Tsc1^ff* embryos (blue) (**d**). (i) Change in the percentages of EGFP+nestin+ NT cells from E15.5 to E18.5 embryos. (ii) Charts representing EGFP+nestin+ NT cells at E18.5. (iii, iv) Change in the percentages of (iii) EGFP+nestin+GFAP+ or (iv) EGFP+GFAP+ NT cells from E15.5 to E18.5 embryos. Data represent means ± s.e.m.. Error bars are defined as means + s.e.m. *$P \leq 0.05$, ***$P \leq 0.001$ *t*-test. Data are representative of two experiments, $n_{GFAP-Cre;ROSAmT/mG}$ (E15.5 $n = 5$, E18.5 $n = 3$), $n_{GFAP-Cre;ROSAmT/mG;Tsc1ff}$ (E15.5 $n = 5$, E18.5 $n = 7$) (**c** (i–iv)); $n_{Nestin-Cre;ROSAmT/mG}$ (E15.5 $n = 4$, E18.5 $n = 5$), $n_{Nestin-Cre;ROSAmT/mG;Tsc1ff}$ (E15.5 $n = 7$, E18.5 $n = 5$) (**d** (i, ii)); $n_{Nestin-Cre;ROSAmT/mG}$ (E15.5 $n = 4$, E18.5 $n = 5$), $n_{Nestin-Cre;ROSAmT/mG;Tsc1ff}$ (E15.5 $n = 6$, E18.5 $n = 5$) (**d** (iii, iv)) embryos

GFAP-Cre driver to remove the floxed *Tsc1* allele from the NSCs and early and late neuronal progenitors, respectively, since both promoters are active in LAM and angiomyolipoma (Fig. 1). We compared control- and mutant-EGFP+ cells: (1) nestin-positive cells, representing NSCs[23, 55]; (2) nestin- and GFAP-expressing cells (radial glial progenitors (RGCs))[23, 55, 56]; (3) nestin- and NS-tubulin-expressing cells (neuronal progenitors)[57]; (4) GFAP-expressing cells (glial precursors); and (5) NS-tubulin-expressing cells (differentiated neurons) (Fig. 5, Supplementary Fig. 7A–C, and Supplementary Table 4).

As embryos matured (E15.5 vs. E18.5), the number of NSCs decreased, as expected, in both control strains (Fig. 5c(i), d(i) and Supplementary Table 4). In *GFAP-Cre;ROSA*$^{mT/mG}$*;Tsc1*$^{ff}$ mice, we observed a slower reduction in the number of *Tsc1*-null NSCs (EGFP$^+$nestin$^+$) (Fig. 5c(i) and Supplementary Table 4), indicated by the higher number, compared to the control (wild-type), of NSCs at E18.5 (Fig. 5c(ii), Supplementary Fig. 7B, and Supplementary Table 4). In *Nestin-Cre;ROSA*$^{mT/mG}$*;Tsc1*$^{ff}$ embryos, the *Tsc1*-null NSCs accumulated with time, with their number tripling by E18.5 (Fig. 5d(i, ii), Supplementary Fig. 7C, and Supplementary Table 4). As an embryo matures, NSCs give rise to the RGCs, which then differentiate along neuronal and glial lineages[58]. Therefore, we examined the rate of differentiation of RGCs and glial precursors.

RGCs (EGFP$^+$nestin$^+$GFAP$^+$) in control mice (*GFAP-Cre; ROSA*$^{mT/mG}$ and *Nestin-Cre;ROSA*$^{mT/mG}$) increased over time by 15- and 60-fold, respectively (Fig. 5c(iii), d(iii) and Supplementary Table 4), indicating the differentiation of wild-type NSCs toward RGCs. However, in *GFAP-Cre;ROSA*$^{mT/mG}$*;Tsc1*$^{ff}$ and *Nestin-Cre; ROSA*$^{mT/mG}$*;Tsc1*$^{ff}$ embryos, RGCs increased by only 3.7- and 1.6-fold, respectively, over time (Fig. 5c(iii), d(iii) and Supplementary Table 4). A similar reduction in differentiation toward glial precursors (EGFP$^+$GFAP$^+$) occurred in the *Tsc-1*-null EGFP$^+$GFAP$^+$ cells, compared to the controls (Fig. 5c(iv), d(iv) and Supplementary Table 4). The slower rate of differentiation along the RGC and glial lineages, in conjunction with the increased number of *Tsc-1*-null NSCs at E18.5, suggests impairment in the *Tsc-1*-null NSC differentiation toward these lineages.

**Oscillation of Hes1 and Rheb blocks NSC differentiation**. Because *Tsc1* loss led to accumulation of NSCs, we next sought to determine whether Tsc1 regulates Notch oscillation and expression of Rheb in NSCs in vivo. The NSCs (nestin$^+$) were isolated from E15.5 *Nestin-Cre;Tsc1*$^{ff}$ and littermate control mouse embryo NTs (Supplementary Fig. 7D, E-i), synchronized by serum starvation, and evaluated for *Rheb* and *Hes1* mRNAs. The analysis of repeated measures demonstrated that *Hes1* oscillation patterns differ significantly ($P = 0.0146$ by regression analysis of longitudinal data) between controls vs. mutant mice, with higher amplitudes of the oscillations in *Tsc1*-null cells. The changes in *Rheb* mRNA expression correlated positively with the *Hes1* oscillations in control (Fig.6a-i, iii, $R^2 = 0.579$, $P < 0.0001$ by regression analysis of longitudinal data) and *Tsc1*-null NSCs (Fig. 6a-ii-iv, $R^2 = 0.36$, $P = 0.02$ by regression analysis of longitudinal data). The mean expression of averaged *Rheb* and *Hes1* was significantly different in *Nestin-Cre;Tsc1*$^{ff}$ embryos vs. wild-type littermate controls ($P = 0.0201$ by regression analysis of longitudinal data). *Gapdh* levels did not change over time as indicated by raw CT values for each time point (Supplementary Fig. 7F). Therefore, we concluded that Tsc1 regulates Notch and, subsequently, Rheb oscillation. WB and densitometry analyses confirmed increased oscillation of Rheb and Hes1 over time in *Tsc1*-null NT cells compared to NT cells of littermate controls (Fig. 6c(i) and Supplementary Fig. 10), which was indicated by 5-fold vs. 1.5-fold increase in the expression of Rheb at 120 min relative to 0 min in NT cells of *Tsc1*-null and littermate control embryos, respectively (Fig. 6c(i) and Supplementary Fig. 10). Furthermore, immunofluorescence (IF) analysis demonstrated higher expression of Hes1 in *Tsc1*-null NT cells and a 'salt-and-pepper' pattern of staining (Supplementary Fig. 7G) indicative for Hes1 oscillation[19]. The partial loss of Notch1 in synchronized E15.5 *Tsc1*-null neuronal progenitors (*Nestin-Cre;Tsc1*$^{ff}$*;Notch*$^{f/+}$) (Supplementary Fig. 7E-ii) reduced significantly ($P = 0.0054$ of averaged *Rheb* and *Hes1* by regression analysis of longitudinal data) oscillation of *Rheb* and *Hes1* mRNAs compared to the wild-type littermate controls, ultimately leading to low and sustained expression of Rheb (Fig. 6b). WB blotting and densitometry analyses confirmed that partial loss of Notch1 function suppresses Rheb and Hes1 oscillation and expression in NT cells (Fig. 6c(ii) and Supplementary Fig. 10). Next, NTs from *Nestin-Cre;*Tsc1$^{ff}$, *Nestin-Cre;Tsc1*$^{ff}$*;Notch*$^{f/+}$, and littermate controls were dissected, dissociated, and subjected to FACS. The NSCs containing pS6 were examined, since pS6 serves as an indicator of mTOR hyperactivation, resulting from the loss of *Tsc1* (Fig. 6d(i)). The loss of Tsc1 increased numbers of pS6$^+$Nestin$^+$NSCs within the NTs of *Nestin-Cre;Tsc1*$^{ff}$ embryos (Fig. 6d(ii)). The partial loss of Notch1 in *Tsc1*-null embryos (*Nestin-Cre;Tsc1*$^{ff}$*;Notch*$^{f/+}$) (Supplementary Fig. 7H) reduced the number of pS6$^+$ cells (Fig. 6d(i)) and pS6$^+$ NSCs (Fig. 6d(ii)), which is consistent with Notch-dependent suppression of Rheb. These results were confirmed using *ROSA*$^{mT/mG}$ mice. The partial loss of Notch1 in *Tsc1*-null embryos (*Nestin-Cre;ROSA*$^{mT/mG}$*;Tsc1*$^{ff}$*;Notch*$^{f/+}$) reduced NSCs when compared to *Tsc1*-null embryos with intact Notch1 alleles (*Nestin-Cre;ROSA*$^{mT/mG}$*;Tsc1*$^{ff}$) (Fig. 6d(iii), Supplementary Fig. 7I). In summary, data support a role of the Rheb-Notch-Rheb loop in maintaining NSCs during mouse embryogenesis because disruption of this loop by removing one allele of Notch suppressed the oscillation of Hes1 and Rheb (Fig. 6b, c(ii) and Supplementary Fig. 10) and prevented the accumulation of NSCs caused by the loss of *Tsc1* (6d(ii, iii)).

**The Rheb/Notch/Rheb loop operates in *Tsc*-null tumors**. We utilized uterine leiomyoma-derived Tsc2-null ELT3 cells[59], since the majority of these cells expressed nestin alone (Fig. 6e(i)). WB confirmed expression of nestin in ELT3 cells (Fig. 6e-ii and Supplementary Fig. 10). We used these cells in a xenograft model[17] to determine the role of the Rheb-Notch-Rheb loop in TSC tumorigenesis. qPCR of the ELT3 tumors revealed a positive correlation between *Rheb* and *Hes1* mRNAs (Supplementary Fig. 7J), which is associated with the high number of nestin-positive cells within these tumors (Fig. 6f(i)-ii, placebo). The experimental disruption of the Rheb-Notch-Rheb loop by DAPT affected the multipotency of the cells because the number of nestin$^+$ *Tsc2*-null cells (Fig. 6f(i, ii)) and *Rheb* mRNA and protein in these tumors (Fig. 6f(iii–v) and Supplementary Fig. 10) were reduced. This was associated with reduction of *Tsc2*-null tumor growth (Supplementary Fig. 7K), consistent with previous reports[17, 18]. DAPT inhibition of Notch signaling was verified by immunoblotting (Fig. 6f(iv) and Supplementary Fig. 10).

**The cell-specific *Tsc1* loss leads to renal carcinoma**. The cell-specific depletion of *Tsc1* in nestin-expressing kidney cells in adult *Nestin-TamCre;Tsc1*$^{f/f}$ mice resulted in multiple bilateral cysts, renal intraepithelial neoplasia (RIN), and invasive multi-focal papillary carcinomas composed of eosinophilic tumor cells (Fig. 7b, c), consistent with morphology observed in TSC-associated RCCs[60]. The origin of renal lesions from nestin-expressing progenitors is suggested by the single nestin-positive cells, containing pS6, which have spindle-shape morphology, similar to these progenitors, within the malignant renal lesions (Fig. 7d–g and Supplementary Fig. 9, arrows: nestin [red], pS6 [green], arrowheads: co-localization of pS6 and nestin). We also determined that renal lesions express Cre recombinase (Fig. 7h) confirming that observed phenotype is linked to Cre recombinase-driven targeting of *Tsc1*. Cre recombinase is expressed at high levels within tumor cells and dysplastic tubules adjacent to papillary neoplastic lesion, indicating that tumor and dysplastic cells origin from Cre-expressing progenitors (Fig. 7h). The less evident expression of Cre in intact glomeruli (Fig. 7h, upper right corner of the image) is expected, as podocytes and the

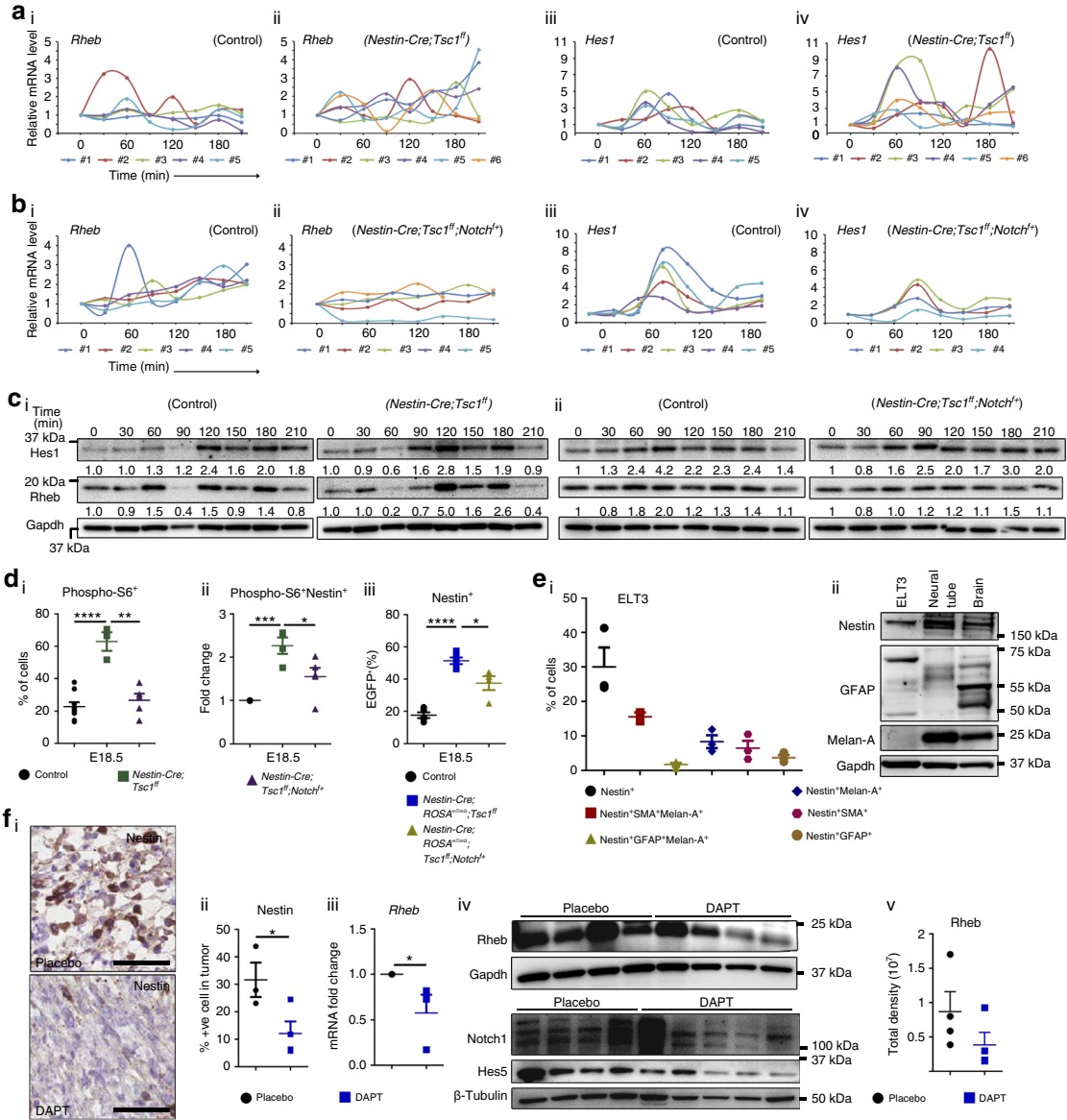

**Fig. 6** Loss of *Tsc1* leads to increased oscillation of Notch and Rheb. **a**, **b** q(RT)-PCR of (i, ii) *Rheb* and (iii, iv) *Hes1* relative to *Gapdh* in NT cells from E15.5 littermate controls or **a** *Nestin-Cre;Tsc1ff* or **b** *Nestin-Cre;Tsc1ff;Notchf+* embryos after release from synchronization. **c** The expression of Rheb and Hes1 in synchronized NT cells from E15.5 littermate controls or (i) *Nestin-Cre;Tsc1ff* or (ii) *Nestin-Cre;Tsc1ff;Notchf+* embryos after release from synchronization by western immunoblotting. Numeric values represent densitometry analysis of the expression of Rheb or Hes1 relative to expression of Gapdh. **d** FACS of (i) phospho-S6 or (ii) phospho-S6 and nestin in NTs. **d** (i) percentage of phospho-S6+ cells in *Nestin-Cre;Tsc1ff* and *Nestin-Cre;Tsc1ff;Notchf+* embryos relative to littermate control embryos (*Tsc1++;Notch++*), at E18.5. (ii) Fold changes in the percentage of phospho-S6+nestin+ NSCs in *Nestin-Cre;Tsc1ff* and *Nestin-Cre; Tsc1ff;Notchf+* embryos relative to littermate control embryos (*Tsc1++;Notch++*) at E18.5. (iii) The percentage of NSCs (EGFP+nestin+) within NTs from *Nestin-Cre;ROSAmT/mG* (control), *Nestin-Cre;ROSAmT/mG;Tsc1ff*, and *Nestin-Cre;ROSAmT/mG;Tsc1ff;Notchf+* embryos at E18.5 (data from *Nestin-Cre;ROSAmT/mG* (control) and *Nestin-Cre;ROSAmT/mG;Tsc1ff* mouse embryos are also used in Fig. 5d(i, ii). **e** (i) Percentages of ELT3 cells expressing nestin alone or co-expressing Melan-A, GFAP, and SMA by FACS. (ii) The expression of nestin, GFAP, and melan-A in ELT3 cells by western immunoblotting. **f** (i) Nestin expression in placebo- (top) and DAPT-treated (bottom panel) xenograft tumors. (ii) Quantification of results shown in **f** (i). (iii) q(RT)-PCR of *Rheb* relative to *Gapdh*. (iv) Rheb, Notch1 and Hes5 in placebo- (n = 6) and DAPT-treated (n = 5) xenograft tumors by western blot. (v) Quantification of Rheb expression shown in **f** (iv) by densitometry analysis after normalization to Gapdh. Data represent means ± s.e.m.. Error bars are defined as means + s.e.m. *P ≤ 0.05' **P ≤ 0.01, ***P ≤ 0.001, ****P ≤ 0.0001 t-test. Data are representative of four, n$_{control}$ = 5 (**a** (i)), n$_{Nestin-Cre;Tsc1ff}$ = 6 (**a** (ii)), n$_{control}$ = 5 (**a** (iii)), n$_{Nestin-Cre;Tsc1ff}$ = 6 (**a** (iv)); three, n$_{control}$ = 5, n$_{Nestin-Cre;Tsc1ff;Notchf+}$ = 5 (**b** (i–iii)), n$_{Nestin-Cre;Tsc1ff;Notchf+}$ = 4 (**b** (iv)); two (**c**); three (**d**, **e**), n = 3 (**d** (i, ii)), n = 4 (**e** (i), **d** (iii)), n = 3 (**f** (ii, iii)), n = 4 (**f** (v)) independent experiments

urinary pole progenitors have smaller cytoplasm compared to large tumor cells and tubules (Fig. 7h). Although the cells that give rise to tumors in Tsc1-null mice are expected to express nestin, only rare tumor cells preserved this expression, likely because of inactivity of the endogenous (mouse) nestin promoter upon differentiation of tumor cells. In summary, our results

support the concept that the loss of *Tsc1* blocks differentiation via the continuously active Rheb-Notch-Rheb loop. This loop activity is maintained by cyclic and continuous binding of Notch1 to the activating NREs on the Rheb and Hes1 promoters. This mechanism facilitates increases in the expression and/or amplitudes of *Rheb* and *Hes1* oscillation (Fig. 7i).

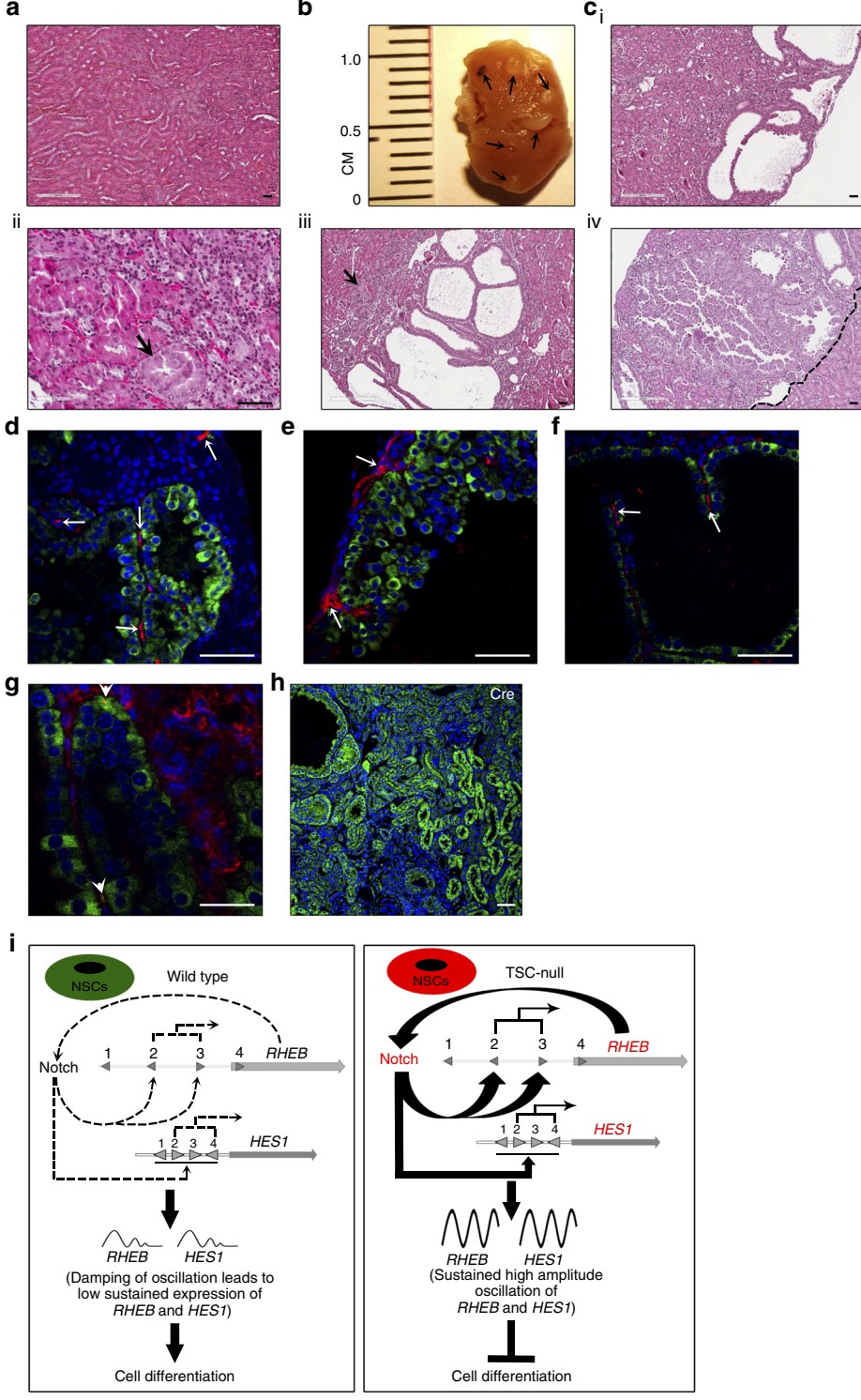

**Fig. 7** Spontaneous renal tumorigenesis induced by cell-type-specific loss of *Tsc1*. **a** H&Es of *Tsc1^{f/f}* (control) kidney, *n* = 3. **b** Kidney lesions in *Nestin-TamCre;Tsc1^{f/f}*; **c** H&Es of *Nestin-TamCre;Tsc1^{f/f}* kidneys, *n* = 3: (i) cysts; (ii) RIN (arrow); (iii) papillary carcinoma (micro-invasion-arrow); (iv) invasive papillary renal cell carcinoma (dashed line indicates border between RCC and normal kidney). Tamoxifen was injected intraperitoneally into 2–3-month-old mice at the dose of 120 mg/kg/day for two consecutive days. Mice were harvested at the age of 7 months. **d**–**f** Expression of nestin (red) in renal spindle-shape tumor cells. Expression of phospho-S6 within renal lesions (green). **g** Co-expression of nestin and phospho-S6 in spindle-shape tumor cells (arrowheads). **h** Expression of Cre recombinase within renal lesions of *Nestin-TamCre;Tsc1^{f/f}* mice. **i** The role of the Rheb-Notch-Rheb loop during NSC maintenance. The loss of Tsc1 blocks the differentiation of NSCs and progenitors via activation of the Rheb-Notch-Rheb loop, which leads to the amplified oscillation of *HES1* and *RHEB*, controlled by the fluctuation in the binding of N1ICD to their activating NREs

## Discussion

We discovered a previously unreported binding of Notch1 to NREs in the promoter of Rheb that regulates Rheb transcription. This binding is a part of the novel regulatory mechanism, which we termed the Rheb-Notch-Rheb loop. We have focused only on NRE2 and NRE3, since the mutation of these sites resulted in significant difference in the activity of Rheb promoter. The NRE2 and NRE3 appear to be regulators for Notch-dependent trans-activation of Rheb. We propose that reduction in activity of NRE2 and NRE3 on the Rheb promoter allows differentiation, since suppression of Rheb via NRE2 and NRE3 becomes apparent during experimentally induced neuronal differentiation. These findings provide insights into mechanisms of aberrant LAM and angiomyolipoma differentiation, since hyperactivation of this loop appears to maintain multipotency of angiomyolipoma cells. Conversely, reduced activity of this loop permits differentiation of angiomyolipoma cells along neuronal lineages. We recognize that the potential RBPJ binding site 1 and NRE2 sites are only 212 bps apart and it may be difficult to distinguish between loading of these by Notch1-ternary complex as a result of their proximity. However, we see consistent and stronger binding of Notch1 to NRE2 and NRE3, but not to the potential RBPJ binding site 1, suggesting that the latter is rather inactive. Our conclusion is supported by suppression of Notch1-ternary complex binding to NRE2 and NRE3 in N-medium and lack of such effect for potential RBPJ binding site 1. The interaction between Notch1 and Rheb might also involve the regulation of Rheb at the protein level. Makovski et al.[61] determined that Rheb protein is stabile for at least 26 h. We observed the significant reduction of Rheb at the protein level within 90 min from release from serum synchronization (Figs. 2b and 6c). Therefore, it is likely that Notch1 also regulates Rheb protein degradation to achieve its oscillatory expression, similar to the regulation of β-catenin protein degradation in stem and progenitor cells[62]. The angiomyolipoma cell line (CRL4004) has only one inactivated and one normal allele of TSC2, and therefore some of data obtained with this cell line should be interpreted with caution. However, both angiomyolipoma cell lines 621−101 and CRL4004 used in this work have multiple common features such as mTORC1 hyperactivation, differentiation abnormalities, binding of Notch1 to the RHEB and HES1 promoters, and oscillation of RHEB and HES1 mRNAs and proteins. Therefore, we consider CRL4004 cell line as an appropriate tool to study angiomyolipoma in addition to well-established 621−101 cells.

The loss of Tsc1 in mouse embryo NTs leads to the accumulation of NSCs, inhibition of differentiation, and increased oscillation of Hes1 and Rheb expression, consistent with our findings in angiomyolipoma cells. This is also consistent with the important function of Notch1 in maintaining high level of Hes1, which retains NSCs in an undifferentiated state. Therefore, the Rheb-Notch-Rheb loop appears to regulate NSC differentiation during embryonic development (Fig. 7i). In addition, we provide evidence for the role of this loop in maintenance of Tsc-null multipotent tumor cells during TSC tumorigenesis.

In summary, we have provided direct and indirect evidence for the significant role of the Rheb-Notch-Rheb loop in maintaining multipotent properties of Tsc-null cells and TSC tumorigenesis, although certain aspects of the proposed model remain to be confirmed. In addition, our data suggest the potential origin for TSC-associated RCC from renal progenitors localized in the Bowman's capsule urinary pole, although more thorough investigation is required to fully support this notion. These data are clinically relevant, since renal cysts are found in 50% of TSC patient and 2–4% of these patients develop TSC-associated papillary RCC[60]. Although Tsc1/2 heterozygosity has been reported to be associated with RCC and cysts in mice[63, 64], our data point to a role of nestin-expressing kidney cells in early-onset RCC.

Our data indicate that Notch1 transactivates Rheb. Rheb mediates the activation of mTOR[20], and mTOR is activated in Notch-dependent malignancies[65], and therefore this study suggests a potential role for the Rheb-Notch-Rheb loop in mTOR activation in non-TSC malignancies without identified TSC1/2 mutations. Therapeutic targeting of this loop might offer a new strategy for patients with Notch/Rheb/mTORC1-dependent tumors.

## Methods

**Human and animal studies.** Human samples were from the National Disease Research Interchange (NDRI) and the Center for LAM Research at Brigham and Women's Hospital with obtained informed consent from all human participants under Institutional Review Board approval.

The following mouse strains were used: B6.Cg-Tg(Nes-cre)1Kln/J, C57BL/6-Tg (Nes-cre/Esr1)1Kuan/J, FVB-Tg(GFAP-cre)25Mes/J, Tsc1tm1Djk/J, B6.129×1-Notch1[tm2Rko]/GridJ, and Gt(ROSA)26Sortm4(ACTB-tdTomato,EGFP)Luo/J (The Jackson Laboratory). Animals, females and males, were on C57BL/6J, FVB/N or mixed C57BL/6J, BALB/cJ, 129/SvJae background. For the inducible promoter tamoxifen dissolved in corn oil (40 mg/ml) was administered intraperitoneally (i.p.; 120 mg/kg/day) for 2 days. Embryos with similar genotype were used in randomized cohorts. Xenograft tumor volume: ELT3 cells were inoculated into the posterior back region of 6-week-old immunodeficient SCID mice (Charles River Laboratories). Tumor length, width, and depth were monitored with a Vernier caliper. When tumors reached 125 mm³, mice were randomly assigned to DAPT (Calbiochem) at the dose of 10 mg/kg/day or vehicle control via i.p. injection for 3 consecutive days, followed by 4-day interval up to 2 weeks[17]. Animal experiments were approved by the Texas Tech University Health Sciences Center Institutional Animal Care and Use Committee according to the National Institutes of Health (NIH) guidelines.

**Cells.** Angiomyolipoma-derived cells: 621−101 with inactivation of both TSC2 alleles[28] (from Dr. Elizabeth Henske), CRL4004 (ATCC) with a 5 bp deletion in TSC2 exon 33[30] from a sporadic angiomyolipoma[29] were cultured in DMEM (high glucose) media (Corning) supplemented with 10% fetal bovine serum (FBS) or heat-inactivated FBS, penicillin/streptomycin, and glutamine[17]. 621−101 and CRL4004 cells were plated in N-medium immediately after trypsynization. N-medium was prepared as follows: DMEM:F12 (Sigma-Aldrich), 15% chicken embryo extract (U.S. Biological Inc. or VWR), N2 and B27 supplements (Life Technology), 20 ng/ml recombinant human fibroblast growth factor-basic (Prospec), 20 ng/ml recombinant human insulin-like growth factor 1, and 20 ng/ml human recombinant epidermal growth factor (R&D System)[44].

Fresh aliquots of HeLa and 293T cells were purchased from ATCC and cells were cultured in DMEM (high glucose) supplemented with 10% FBS, penicillin/streptomycin, and glutamine.

SK-Mel2, SK-Mel5, and Malme (all from ATCC) were maintain in RPMI supplemented with 10% heat-inactivated FBS, penicillin/streptomycin, and glutamine. ELT3 cells (from Dr. Elizabeth Henske) were maintained in IIA complete medium (50% DMEM:F12, 1.2 g/L NaHCO3, 1.6 μM FeSO4, 50 nM sodium selenite, 25 μg/ml insulin, 200 nM hydrocortisone, 10 μg/ml transferrin, 1 nM triiodothyronine, 10 μg/ml vasopressin, 10 nM cholesterol, 10 ng/ml epidermal growth factor, 15% FBS, and penicillin/streptomycin[59]). Cells were treated with 1 nM rapamycin (Biomol) or dimethyl sulfoxide. Cells identity was verified by the STR-DNA profiling (Genetica). All cell lines were routinely tested for mycoplasma contamination (InvivoGen). Stable lines were generated by infection with lentiviruses packaged with psPAX2 and pMD2.G vectors containing control, Rheb shRNA (Addgene)[46], or Notch shRNA (Sigma) and selected with 8 μg/ml puromycin[66].

**Mutational analysis of CRL4004.** The comprehensive TSC1/TSC2 genetic analysis was performed by massively parallel sequencing using a customized gene bait set, yielding 500× coverage of the entire genomic extent of TSC1 and TSC2. This analysis confirmed the known frameshift TSC2 mutation c.4081_4085delCGAGT/ p.V1362Lfs* with allele frequency 48% (hg19) (previously reported by Lim et al.[30]) in the CRL4004 cell line. Allele frequency single-nucleotide polymorphism analysis in TSC1/TSC2 genes showed no hint for loss of heterozygosity (LOH) in CRL4004 cell line (skewed allele frequency AF< 0.4 or AF > 0.6). This highly sensitive analysis strongly suggests that the CRL4004 cell line has one inactivated and one normal allele of TSC2.

**Western blotting.** Cells were lysed using RIPA, Tris-Triton buffer (for an intermediate filament, microtubules, or cytoskeletal proteins) (10 mM Tris, pH 7.4, 100 mM NaCl, 1 mM EDTA, 1%Triton X-100, 10% glycerol, 0.1% SDS, 0.5% deoxycholate) or nuclear extraction buffer (20 mM HEPES, pH 7.9, with 1.5 mM MgCl, 0.42 M NaCl, 0.2 mM EDTA, and 25% (v/v) glycerol). Then, 5 μl of 0.1 M

dithiothreitol solution and protease and phosphatase inhibitor cocktails were added to 500 μl of 1× lysis buffer. Lysates were subjected to electrophoresis (Invitrogen).

**Immunoassays**. The following antibodies were used: Rheb (Cell Signaling #13879, 1:1000), NSE (Abcam #ab16808, 1:1000), β-3-tubulin (Abcam #ab78078, 1:200 (IF), 1:5000 (WB)), nestin (Abcam #ab6142, 1:100 (IF)), peripherin (Abcam # ab4666, 1:1000), GFAP (Sigma-Aldrich # G9269, 1:200 (immunohistochemistry (IHC)), 1:1000 or 1:5000 (WB)), β-actin (Sigma-Aldrich A5316, 1:2000), β-tubulin (Sigma-Aldrich # T8328, 1:5000) c-myc (Santa Cruz Biotechnology #sc-40, 1:500), Notch1 (C-20) (Santa Cruz Biotechnology #sc-6014-R, 1:500), Hes5 (Santa Cruz Biotechnology #13859, 1:500); NSE (BioGenex, # MU055-UC, 1:50 (IHC)); nestin (Novus Biologicals # NBP1–02419 1:1000); cleaved Notch1 (Val1744) (Cell Signaling #4147, 1:1000), hamartin (Cell Signaling #4906, 1:1000), phospho-S6 (Cell Signaling #2211, 1:1000 (WB), 1:400 (IF)), hes1 (Cell Signaling #11988, 1:1000 (WB)), hes1 (Genetex #GTX62458, 1:200 (IF)), phospho-S6K (T389) (Cell Signaling #9202, 9206, 1:1000) and Gapdh (Cell Signaling #5174P, 1:3000); phospho-S6-PacificBlue (Cell Signaling #8520, 1:320), nestin-PE/AF647 (BD Biosciences #561230, 1:100), GFAP-PE/AF647 (BD Biosciences #561470, 1:40), β-3-tubulin-FITC (Sigma-Aldrich #SAB4700545, 1:100), melan-A (Thermo Scientific #MA5-15237, 1:500), SMA-PerCP (R&D Systems #IC1420C, 1:10), Ki-67-PE/Cy7 (BD Biosciences #561283, 1:80); Cre recombinase (EMD Millipore # MAB3120, 1:500 (IF)). Secondary steps for IHC were performed using detection kit (Invitrogen). Images were captured with Nikon TE2000 microscope. FACS events were acquired using LSRFortessa (BD Bioscience).

**IHC quantification using Aperio**. Aperio software system removes all the background threshold prior to analysis, and therefore the area of co-localization is free of background error. Aperio digital pathology systems uses algorithm-based software analysis to locate and quantitate the staining. The co-localization algorithm separates the stains based on the RGB (red, green and blue) values into three channels corresponding to the colors used, allowing quantifying the area and intensities of each stain separately and the co-localized markers. After running the algorithm analysis, the software generates a 'Markup image' with an arbitrary color for each stain and for areas where stains co-localize. The negative areas with only hematoxylin are marked in Blue, Red and Green for individual stained areas and Yellow in the areas where the stains superimpose. The yellow color is arbitrary choice by the Aperio software to indicate areas of co-localization and it does not reflect the color resulting from mixing two different colors as seen during immunofluorescence detection.

**Transfection and reporter gene assays**. Human MAML1 and N1ICD were cloned into pCS2 containing an N-terminal 6X-MycTag using Gateway technology and standard PCR. Rheb luciferase and control promoter constructs were purchased from Lightswitch Genomics (catalog number S717541). Hela cells were seeded at 20,000 cells per well in 96-well plate format. The next day, the media were changed to Opti-MEM I and cells were transiently transfected with 75 ng of the Rheb-Promoter construct, 10 ng of pGL3-Control, 50 ng of N1ICD, and 50 ng of MAML1 and 35 ng of pCS2 using Lipofectamie 2000 transfection reagent (Invitrogen). Cells were harvested 48 h later in 50 μl of 1× Passive Lysis Buffer and the levels of luciferase were measured with the Dual Luciferase Assay System from Promega on a Modulus II Luminometer with 10 s integration time per luciferase assay and 2 s delay before taking the reading. The readings were obtained in Microsoft Excel software. The measurement readings were given as relative light units, and analyzed as a ratio of the *Renilla* luciferase measurements to the control Firefly luciferase measurements.

**Site-directed mutagenesis**. The Rheb-NRE mutants were generated as previously described[43] using QuikChange (Stratagene) and the following primers: NRE1-MUT-S-5′CAGACTCCATGCTTGCGACAGTCCTTCGGCC-3′/NRE1-MUT-AS-5′-GGCCGAAGGACTGTCGCAAGCATGGAGTCTG-3′, NRE2-MUT-S-5′CAA TCACCGCACCTGCGACCTATTGCCCCGC-3′/NRE2-MUT-AS-5′-GCGGGG CAATAGGTCGCAGGTGCGGTGATTG-3′, NRE3-MUT-S-5′-CCTTGTTTC CCCCAATCGCAGATGGAGTTTCCG-3′/NRE3-MUT-AS-5′-CGGAAACTC-CATCTGCGATTGGGGGAAACAAGG-3′, NRE4-MUT-S-5′-AAAGCGGCG-GAAGAAGGTCGCAGGGTCATGAC-3′/NRE4-MUT-AS-5′- GTCATGACC CTGCGACCTTCTTCCGCCGCTTT-3′. For each mutant, Sanger sequencing (Genewiz) verified the integrity of the whole promoter region and base pair changes.

**Genome editing with CRISPR Cas9**. The single-guide RNAs (sgRNAs) targeting genomic regions of interest were designed using CRISPR Design Tool (http://portals.broadinstitute.org/gpp/public/) and the guide sequences are as follows: NRE3 G2′-S-5′-CACCGAGCTGTCCAATCGGCGCTCG-3′/NRE3 G2′-AS-5′-AA ACCGAGCGCCGATTGGACAGCTC-3′/NRE3-G3′-S-5′-CACCGGTGTATT TTTAGCTCCCGGG-3′/NRE3-G3′-AS-5′-AAACCCCGGGAGCTAAAAATA-CACC-3′. The sgRNA oligos were subcloned into pSpCas9(BB)-2A-GFP (PX458) Addgene vector and transiently transfected into CRL4004 cells. At 48 h post transfection, cells were sorted using FACS Aria (BD) and plated to 96-wells plate at a density of 0.8 cell per well. Cell grown up to 90% confluency underwent

subculture for maintenance, genomic purification and PCR, and western immunoblotting analyses. For PCR amplification of NRE3 region, NRE3-S-5′-CGGT AGCAGCGAGGTGTATT-3′ and NRE4-AS-5′-GTCGGGGCGACGTTTTACTT-3′ primers were used and clones were screened. Amplicons of 272 bp indicated intact DNA. Amplicons of 165 bp indicated clones with prospective focal deletion of NRE3 region and those were subjected for sequencing.

Neural tube dissociation was performed using papain according to the manufacturer (Worthington).

**Real time quantitative reverse transcription**. The q(RT)-PCR was performed using High Capacity cDNA Synthesis Kit, Fast SybrGreen, StepOnePlus Applied Biosystems and the following primers: *SOX10*-F-5′-ATGAACGCCTTCATGG TGTGGG-3′/R-5′-CGCTTGTCACTTTCGTTCAGCAG-3′; *NGN1*-F-5′-CCCCT AGTCAGCAGGCAATA-3′/R-5′-CCTAACAAGCGGCTCAGGTA-3′; *S100A1*-F-5′-TGGACTTCCAGGAGTATGTGG-3′/R-5′-TGCTCAACTGTTCTCCCAGA-3′; *DCT*-F-5′-CCTAGGGTGCTCATGCCTTA-3′/R-5′-CAACTCAAGAAGGAACA GTGAGG-3′; *c-KIT*-F-5′-GCAAATACACGTGCACCAAC-3′/R-5′-GCACCCCTT GAGGGAATAAT-3′; *ID1*-F-5′CCAACGCGCCTCGCCGGATC-3′/R-5′-CTCCT CGCCAGTGCCTCAG-3′; *ID3*-F-5′-CAGCTTAGCCAGGTGGAAATCC-3′/R-5′-GTCGTTGGAGATGACAAGTTCCG-3′; *SOX9*-F-5′-AGGAAGCTCGCGGAC-CAGTAC-3′/R-5′-GGTGGTCCTTCTTGCTGCAC-3′; *NOTCH1* and *RHEB* (human and mouse, both from Qiagen or Applied Biosystem/TaqMan); *Rheb* (rat)-F-5′-CTGACCAGGCTACCAAGATG-3′/R-5′-CAATGAGGACTTTCCCACAGA-3′; *HES1*-F-5′-GGAAATGACAGTGAAGCACCTCC/R-5′- GAAGCGGGTCAC CTCGTTCATG-3′; *Hes1*-(from Applied Biosystem/TaqMan)'; *Hes1*-(rat)-F-5′-CAACACGACACCGGACAAAC-3′/ R-5′-GGAATGCCGGGAGCTATCTT -3′; *GAPDH*- F-5′-ACTGACACGTTGGCAGTGG-3′/ R-5′-GGCTCTCCAGAA-CATCATCC-3′; *Gapdh*-(mouse)-(from Applied Biosystem/TaqMan); *Gapdh*-(rat)-F-5′-GACATGCCGCCTGGAGAAAC-3′/R-5′-AGCCCAGGATGCCCTTT AGT-3′; 18S-F-5′-GTCTGTGATGCCCTTAGATG-3′/R-5′-AGCTTATGACC CGCACTTAC-3′.

Relative expression was calculated using the 2-ΔΔCt method and RT2 profiler PCR Array Data Analysis (SAB Biosciences) and normalized to GAPDH.

**Chromatin immunoprecipitation quantitative PCR**. ChIP-qPCR was performed using EZ-ChIP (Millipore) or SimpleChIP Enzymatic Chromatin IP kit (Cell Signaling). Immunoprecipitation was performed using α-Notch1 (Abcam) or IgG control. qPCR for Rheb-NREs was performed using primers: NRE-1-S-5′-CG AGGCTCGCCCTGTTTTTA-3′/NRE-1-AS-5′-CTTGAGCTGCGTTAGCGTTC-3′, NRE2-S-5′-ACCCAGACACGAACGCTAAC-3′/NRE2-AS-5′-GAGAGACAA ACGGTGGCTCC-3′; NRE-3-S-5′-CGGTAGCAGCGAGGTGTATT-3′/NRE-3-AS-5′-GATTGGACAGCTCCTGCACA-3′. For Notch binding to Hes1 promoter the following primers were used: HES1-S-5′-GCGTGTCTCCTCCTCCCATT-3′/HES1-AS-5′-CCTGGCGGCCTCTATATATA-3′[67].

**RNA sequencing analysis**. The preprocessed RNA-Seq data were imported into the Partek Flow (Partek, CA). The genome versions of Homo sapiens hg19 and STAR aligner were used for alignment and alignment algorithm. Quantification and normalization were performed with an expectation maximization algorithm. Genes displaying differential expression between normal kidney from patient with sporadic angiomyolipoma, angiomyolipoma tumor, and two angiomyolipoma-derived cell lines (621−101 and CRL4004) were detected using poisson model, assuming that the variance is equal to the mean. The *P*-value is obtained using Wald. *P*-values were adjusted to account for multiple testing using the false discovery rate (FDR) method of Storey and Tibshirani. Only genes that were significantly (FDR < 0.05) modulated in angiomyolipoma tumor and two angiomyolipoma-derived cell lines (621−101 and CRL4004) compared with normal kidney were considered for further analysis.

To assess biological relationships among genes, we used the Ingenuity Pathway Analysis software (QIAGEN Inc.). Network analyses, functional analyses, and canonical pathway analyses were generated.

**Statistical analysis**. Kolmogorov–Smirnov, unpaired *t*-tests, one-way or two-way analysis of variance (ANOVA), non-parametric Mann–Whitney tests, or the regression analysis of longitudinal data (repeated measures) were used. Standard error of means (s.e.m.) was estimated. The variance was similar between groups that were compared. For normalized q(RT)-PCR, fold changes, FACS, and densitometry analyses one-sample *t*-tests were used. Outliers were identified using Grubb's test at α = 0.05. Analyses were done using GraphPad Prism 6 and Statistica.

Based on power calculation using the non-central *t*-distribution for a two-sided *t*-test, we have estimated that analysis of 10 mice per cohort allow us to achieve an actual power of 75% power to distinguish an effect, defined as the difference in means relative to the standard deviation, of 1.25 between two treatments at the 5% level of significance. However, the preliminary studies have demonstrated that the inclusion of 5 subjects per cohort is sufficient. For animal studies analyses were performed by researcher blinded to sample identity.

**Data availability**. The authors declare that data supporting the findings of this study are available within the paper and its supplementary information files. All additional relevant data are available from the authors.

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

## Acknowledgements

We acknowledge Aristotelis Astreinidis for comments, Deborah McClellan for editorial assistance, Maureen Murphy for providing H1299 cells, Navin Chintala, Sharad Sharma, and Kelly Hartley for technical assistance, the Developmental Corporation of Abilene for support, and the use of tissues procured by the NDRI supported by NIH 2 U42 OD011158. This work was supported by the National Institute of Health (R01CA190209 to M.M.M), the National Science Fundation (1052039 to B.W.), the Cancer Prevention and Research Institute of Texas (RP120168 to M.K.), the U.S. Department of Defense (TS140010 to M.K.), Laura W. Bush Institute for Women's Health (seed grants to M.M.M and M.K.).

## Author contributions

Conception and design: J-H.C., E.P.H., F.R., M.M.M., B.W. and M.K. Development of methodology: J-H.C., B.P., S.B., S.M., M.P., S.G, F.R., K.G., D.J.K., B.W., M.M.M. and M. K. Acquisition of data: J-H.C., B.P., S.B., S.M., S.G., M.P., S.K.V., J.P., E.P.H., F.R., K.G., D.J.K., M.M.M. and M.K. Analysis and interpretation of data: J-H.C., B.P., S.B., S. M., Y.Z., M.P., S.G., E.P.H., F.R., K.G., D.J.K., H.M., M.M.M., B. W., and M.K. Writing and review of the manuscript: J-H.C., B.P., S.M., Y.Z., E.P.H., F.R., K.G., D.J.K., H.M., M. M.M., B.W. and M.K. Study supervision: M.M.M., B.W. and M.K.

## Additional information

**Competing interests:** The authors declare no competing financial interests.

