## [Peer review file · Nature Communications]

Reviewers' comments:

Reviewer #1 (Remarks to the Author):

Cho et al. demonstrated that Notch directly binds to the promoter of Rheb and stimulates its gene expression. The authors identified several Notch-responsive elements (NREs) within Rheb promoter. Intriguingly, the binding of Notch to the NRE2, one of the NREs, of Rheb promoter induces the oscillatory transactivation of Rheb gene as seen in other Notch's targets such as Hes1. They demonstrated that this Notch-induced Rheb activation plays an important role in amplifying the Notch signaling itself, which is termed as the Rheb-Notch-Rheb regulatory loop in this paper. Constitutive activation of Rheb in cells lacking the functional TSC complex leads to aberrant activation of the Rheb-Notch-Rheb regulatory loop and blocks cell differentiation, which may underlie the mechanisms of TSC tumorigenesis.

This is an interesting paper with large amount of data. The most important observation in this paper would be the identification of Notch1 as a direct activator or suppressor of Rheb gene expression. Overall, data are clean and convincing. However, it remains unclear whether the binding of Notch to the promoter of Rheb is indeed an essential event for the blockade of cell differentiation and tumor formation in TSC null cells. This point should be directly examined. In addition, there are a few questions that should be addressed by the authors to confirm the role of NRE2 in the regulation of Notch1-dependent transactivation of Rheb, and the origin of TSC tumor in kidney.

Comments:

1. From the data in Fig. 2C and 2D, the authors concluded that Notch1 stimulated Rheb gene expression through its interaction with the NRE2 of Rheb promoter. The activity of the wild type Rheb promoter was inhibited by just 50% with the treatment of DAPT (Fig. 2D), suggesting that the promoter has a basal activity independent of the N11CD. This is consistent with the data shown in Fig. 2C that the wild type Rheb promoter has relatively high basal activity without detectable endogenous N11CD (lane 1, Fig. 2C). However, mutations in the NRE2 almost completely killed the activity of Rheb promoter, raising the possibility that the NRE2 may have a fundamental role for the activation of Rheb promoter even in the absence of the input from the Notch signaling. The authors should exclude this possibility by replacing the NRE2 with NRE1 sequences and/or other well-known NREs, and confirm the binding of N11CD to the replaced NRE2 and the N11CD-induced activation of its promoter without affecting their basal promoter activity.
2. As mentioned above, it remains unclear whether the binding of N11CD to the NRE1 and/or the NRE2 plays a key role in controlling cell differentiation and/or multipotency. It is important to mutate endogenous NRE1 and/or NRE2 in AML cells to examine their differentiation.
3. The authors stated that deletion of TSC1 in nestin expressing podocytes resulted in the formation of cysts, RIN, and renal cancer. However, the data in Fig. 7 show that the lesions are likely to be originated from tubular cells, not podocytes. There are many intact glomeruli surrounding the lesions. In addition, renal tubular cell-specific TSC1 knockdown but not podocyte-specific TSC1 knockout causes the aberrant cystic formation. The authors should demonstrate TSC1 ablation in podocytes in nestin-cre mice by staining p-S6 or carefully discuss the origin of the cysts and tumors in these mice.

Reviewer #2 (Remarks to the Author):

TSC-RHEB-NOTCH ms rev

0. Since there is no evidence that any of the data generated is of a normal distribution, and consistent with full presentation of all data, please add/replace all graphs that show mean and SE marks with dot plots showing all data points. E.g. 1Dii, 1F, 1G, 2A,

1. Figure 1Dii. the % of co-expression seems quite low, 0.2-6%. Is this significantly above background for two randomly chosen proteins assessed by IHC? I don't see any yellow in 1Di 'Markup' only some green. How does red + brown give either yellow or green?
2. Figure 2A legend states: Data are representative of three, n=3 (A-iii-vi, C, D). Please average together all of the data points and show in the figure rather than just a representative set. Are the differences at the different times statistically significant? Do GAPDH levels change over time? This could explain differences in RHEB and Hes1.
3. What is the number of NREs in the human genome, and the distribution of such elements seen within 5 kb of the TSS for all human genes?
4. The binding shown in Figure 2Bii seems quite modest. Nonexistent for NRE1. Please perform this expt for other validated NOTCH binding sequences in other genes, and compare with the enrichment seen here. How can NRE1 ablation as in Ci have an effect on RHEB expression when it does not bind NOTCH, as in Bii?
5. The methods do not provide sufficient details on the constructs and methods shown in Figure 2Ci and ii.
6. Why is the expression of RHEB reduced in the N1ICD+MAML1 expression experiment, c/w N1ICD alone in figure 2Ci, whereas Ci shows enhanced activity of the promoter with both expressed?
7. For Figure 2D, please show a control to demonstrate effects of DAPT on a canonical NOTCH-NRE promoter.
8. Figure 2Ei, the 210 data point for NRE1 does not look statistically significant. Again, dot plots should be shown for this data, and would be very informative.
9. The 621 cells are well-known and well-characterized. The CRL4004 and CRL4008 cell lines, on the other hand, are not. Please provide documentation that these cell lines are derived from angiomyolipoma: 1) show biallelic mutation/inactivation of TSC2, 2) show by immunoblotting that they do not express TSC2, and have unregulated activation of mTORC1.
10. Why do NS-Tubulin levels rise to ~40% on day 7 in 3Cii, yet only ~7% in Dii?
11. The lack of RHEB and Hes1 cycling in Figure 4Ai is expected since the cells are not going through a serum-deprive-addback treatment, and can't expect this cycling to occur.
12. The observations in Figure 4, and corresponding text are interesting but observational only, and it is likely that many changes in many signaling pathways occur with the switch to N-medium. To connect this to the NOTCH-RHEB pathway directly, does overexpression of RHEB block the phenotypes seen in N-medium?
13. Figure 1ABCD. It would be nice to have quantitative information on the expression of all of these neural-related antigens than just IHC. Please add figures showing expression of them all by immunoblotting, similar to what is seen in Cii.
14. Note that many previous studies have shown differentiation abnormalities in neural precursor cells lacking Tsc1/Tsc2. *Mol Cell Neurosci.* 2002 Dec;21(4):561-74. is one of the earliest.
15. What is the control in Figure 6Aii? It looks very different from the control in 6Ai. There is a comment about strain background effects, but that is not credible for the difference. Please show data for Hes1 in the Nestin-cre Tsc1^{ff} Notch^{f+} cells.
16. Please show immunoblot data for pS6, Rheb, Tsc1, Tsc2, mTORC1 signaling for the embryo cell cultures shown in Figure 6.
17. Show blot data for all the proteins analyzed in 6C on the ELT3 cells to confirm there is an increase in expression.
18. The authors ignore *J Clin Invest.* 2010 Jan;120(1):103-14. doi: 10.1172/JCI37964, which also showed a role for Notch in TSC signaling, and a response to DAPT treatment.
19. Figure 7 does not indicate the age of the mice at the time of injection with tamoxifen, or the age of the mice at the time of sacrifice.
20. What was the survival of all of the brain model mice? Was this significantly improved by presence of the Notch1^f allele?

21. The mouse kidney model data (Figure 7) could have occurred due to off-target expression of cre in any kidney epithelial cell. The pathology is similar/identical to that seen in multiple kidney models over the past 15 years including the Tsc2^{+/+} and Tsc1^{+/+} mice.

Reviewer #3 (Remarks to the Author):

Cho et al. propose the existence of a Rheb-Notch-Rheb autoregulatory loop that regulates the differentiation of neural crest derived progenitors and is instrumental in generating phenotypes associated with tuberous sclerosis. These studies follow up on prior work from the PI suggesting the TSC/Rheb act upstream of Notch in Tuberous sclerosis and in *Drosophila*. This is an intriguing hypothesis, but much of the data in the current manuscript as it relates to Notch is problematic.

1. Figure 1 selects a handful of markers, some not specific such as NSE, to back up the idea that AML cell lines are plastic with respect to differentiation. What does you see if you use an unbiased approach (e.g., RNAseq or expression profiling)?

2. The reporter experiments performed in 293T cells are problematic for several reasons. These cells express viral E1A 13S and E1A 12S proteins that bind and variously activate or inhibit RBPJ-dependent genes in a completely Notch independent fashion, confounding clear interpretation of the results of Notch reporter gene assays (Ansieau et al. *Genes & Development* 15:380, 2001). 293T cells also drive very high expression of transfected genes if plasmids with large T origins of replication are used, raising questions about the relevance of the observed modest activation of the Rheb promoter element in the experimental data provided. Additional experiments in another heterologous cell type are needed to confirm conclusions drawn from the 293 cell experiments.

3. The idea that NRE1 binds Notch1 and is inhibitory is inconsistent with all past data regarding Notch transcription complexes and transcription; there are no well-characterized examples of Notch complexes having repressive effects. Moreover, to truly prove that NRE1 and NRE2 are involved in Rheb regulation in the appropriate cellular context would require other approaches, such as CRISPR mutagenesis putative Notch binding elements in AML lines.

4. Nowhere do the authors actually measure activated Notch1, since they rely on a Santa Cruz antibody reagent that recognizes the intracellular C-terminus of Notch1. Many of the western blot analyses need to be redone with a NICD1 specific antibody, such as that available from Cell Signaling Technologies.

5. Is Rheb really oscillating? There are some modest inconsistencies in the upward trend of Rheb expression after stimulation in DMEM, but there are no nadirs akin to what is seen with Hes1, which truly does oscillate, except in one experiment (Fig. 6Aiii).

6. The authors use qChIP-PCR to show loading of Notch1 onto NRE2 and not NRE1 at most time points following cell stimulation (Fig. 4Bi and Bii). These two sites are only 212 bps apart. What is the size of the chromatin-DNA fragments used in these experiments? It might normally be difficult to distinguish between loading of sites that are this close together.

7. With respect to changes in Notch as they relate to neuronal differentiation (Figure 4D), the data suggest that Notch1 expression is downregulated as the cells adopt a neuronal fate. Notch signaling in other contexts has been reported to be important in upregulating mTOR and S6 phosphorylation as well as cell growth. Hence, a simpler explanation for the observed changes in phenotype is that the N-medium downregulates Notch1 expression through currently unknown mechanisms, leading to downstream effects that include loss of mTOR activity. Rheb knockdown may convey a similar phenotype by also interfering with mTOR and its downstream effects. Stated another way, the more complex Rheb-Notch-Rheb signaling axis proposed by the authors and the

need for oscillatory loading of Notch1 onto Rheb may be an unnecessary complication.

8. The spontaneous renal tumorigenesis shown in figure 7 is impressive, but does not address the proposed model. It would be of interest to know if, for example, conditional haploinsufficiency for Notch1 suppressed or modified tumorigenesis in the mouse.

We thank the reviewers for positive comments and constructive criticism. Please find our responses to reviewers' comments below in blue. The figures marked with Roman numerals are only included in the rebuttal. The remaining numbers refer to the revised figures in the manuscript.

Reviewer #1 (Remarks to the Author):

Cho et al. demonstrated that Notch directly binds to the promoter of Rheb and stimulates its gene expression. The authors identified several Notch-responsive elements (NREs) within Rheb promoter. Intriguingly, the binding of Notch to the NRE2, one of the NREs, of Rheb promoter induces the oscillatory transactivation of Rheb gene as seen in other Notch's targets such as Hes1. They demonstrated that this Notch-induced Rheb activation plays an important role in amplifying the Notch signaling itself, which is termed as the Rheb-Notch-Rheb regulatory loop in this paper. Constitutive activation of Rheb in cells lacking the functional TSC complex leads to aberrant activation of the Rheb-Notch-Rheb regulatory loop and blocks cell differentiation, which may underlie the mechanisms of TSC tumorigenesis.

This is an interesting paper with large amount of data. The most important observation in this paper would be the identification of Notch1 as a direct activator or suppressor of Rheb gene expression. Overall, data are clean and convincing. However, it remains unclear whether the binding of Notch to the promoter of Rheb is indeed an essential event for the blockade of cell differentiation and tumor formation in TSC null cells. This point should be directly examined. In addition, there are a few questions that should be addressed by the authors to confirm the role of NRE2 in the regulation of Notch1-dependent transactivation of Rheb, and the origin of TSC tumor in kidney.

Comments:

1. From the data in Fig. 2C and 2D, the authors concluded that Notch1 stimulated Rheb gene expression through its interaction with the NRE2 of Rheb promoter. The activity of the wild type Rheb promoter was inhibited by just 50% with the treatment of DAPT (Fig. 2D), suggesting that the promoter has a basal activity independent of the N1ICD. This is consistent with the data shown in Fig. 2C that the wild type Rheb promoter has relatively high basal activity without detectable endogenous N1ICD (lane 1, Fig. 2C). However, mutations in the NRE2 almost completely killed the activity of Rheb promoter, raising the possibility that the NRE2 may have a fundamental role for the activation of Rheb promoter even in the absence of the input from the Notch signaling. The authors should exclude this possibility by replacing the NRE2 with NRE1 sequences and/or other well-known NREs, and confirm the binding of N1ICD to the replaced NRE2 and the N1ICD-induced activation of its promoter without affecting their basal promoter activity.

Upon activation of Notch1 receptor, N1ICD assembles with MAML1, and recruits CDK8 and RNAPII, through Mediator to form Notch1 ternary complex. This complex binds to CBF1 that occupies NRE sites. Thus, N1ICD does not bind directly to NREs, but to CBF1 that occupies NREs. CBF1 is a potent DNA-binding transcription factor that exchanges repressors for activators in the response to N1ICD and Notch ternary complex binding. There is no published evidence that NREs regulate any promoter activity without N1ICD participation. In addition, the difference between NRE1 and NRE2 is a single nucleotide (NRE1: TCCCAC and NRE2: CCCCAC) and there is no evidence in the literature that this nucleotide is important for the activation of target genes. Conversely, it has been previously demonstrated that this nucleotide is not required for the binding of CBF1 to NREs (Wilson and Kovall; EMBO, 23, 3441-3451, figure 7). Therefore, replacing the NRE2 with NRE1 sequences has limited potential to clarify if NRE2 regulates activity of Rheb promoter without N1ICD participation.

To address the reviewer's concern, we tested activity of Rheb luciferase promoter using a different cell system developed in the laboratory of Dr. White (described in details in the Material and Method section). Using this system and HeLa cells, we directly demonstrated Notch-dependent regulation of the activity of Rheb promoter. We found that two NRE sites, NRE2 and NRE3, are equally involved in regulation of Rheb promoter

(Fig. 2D-i). The activity of NRE1 and NRE4 mutants was similar to the wild type Rheb, while the basal activity of NRE2 and NRE3 mutants was reduced by 2-fold (Fig.2D-i). These new results suggest that both, NRE2 and NRE3, regulate Rheb expression and that they are responsible for approximately fifty percent of basal activity of Rheb promoter.

To support further Notch1-dependent regulation of Rheb, we also examined whether overexpression of N1ICD and MAML1 affects the expression of endogenous Rheb. In all tested cell lines, we observed the consistent increase in expression of Rheb (Fig.2D-ii). The numeric values in this figure represent the densitometry analysis of expression of Rheb relative to expression of Gapdh.

Conversely, lowering expression of Notch1, using validated Notch shRNA (Sigma), reduced expression of endogenous Rheb, Notch1 and Hes1 (Fig. 2D-iii-iv). The reduced expression of Rheb at the protein level was associated with the reduced levels of Rheb mRNA.

Together these data support the role of Notch1 in regulating Rheb expression and suggest that Notch1-dependent expression of Rheb contributes to approximately fifty percent of the total Rheb promoter activity. This conclusion is included in the revised manuscript.

The lack of strong activation of the Rheb promoter by Notch1 and Mastermind is consistent with the results from Dr. White, who tested activation of promoters of other genes (TWIST1, cMyc, etc) by Notch 1. These promoters DO NOT have a paired site (called a Sequence Paired Site or SPS in the literature). We suspect that differences in performing the reporter gene assays may also account for a difference in strength of promoter activation. In addition, the episomal vs. chromosomal integration of plasmids impacts the strength of promoters' activation evaluated through the reporter assays as discussed by Inoue et. Al [Genome Res. 2017 Jan;27(1):38-52. doi: 10.1101/gr.212092.116. Epub 2016 Nov 9. PMID: 27831498]. All these factors provide plausible explanation for two-fold activation of Rheb reporter vs. seven-fold activation of Hes1 reporter by N1ICD and MAML1, the latter was used as a positive control for Rheb luciferase reporter assay in Dr. White laboratory, and it is not included in this manuscript.

2. As mentioned above, it remains unclear whether the binding of N1ICD to the NRE1 and/or the NRE2 plays a key role in controlling cell differentiation and/or multipotency. It is important to mutate endogenous NRE1 and/or NRE2 in AML cells to examine their differentiation.

We agree that mutating endogenous NRE2 and NRE3 in angiomyolipoma cells, followed by examining cell differentiation would be optimal to determine roles of NRE2/3 in this process. Therefore, we attempted to remove NRE2/3 sites from the promoter of endogenous Rheb in angiomyolipoma cells using CRISPR approach. However, despite our multiple efforts, described in details below, we failed to achieve this goal

Approach to remove NRE2 is shown.

The NRE2 guides were design using broad CRISPR designer tool at <http://portals.broadinstitute.org/gpp/public/> (Fig. I).

The eleven combinations of two guides (described below) were transfected to the 293T and angiomyolipoma cells. The guides' sequences are as follow:

- 1/ NRE2 G1-S 5'-CACCG ccaaccgcttagcgtttcgc-3'/NRE2 G1-AS 5'AAAC gcgaaacgctaagcgggtggC-3'
- 2/ NRE2 G4-S 5'-CACCG cttagcgtttcgccaatcac-3'/NRE2 G4-AS 5'AAAC gtgattggcgaacgctaagC-3'
- 3/ NRE2 G10-S 5'-CACCG gcgctccggttctggccaatc-3'/NRE2 G10-AS 5'AAAC gattggccagaacggagcgcC-3'
- 4/ NRE2 G36-S 5'-CACCG atcaccgcacctcccaccta-3'/NRE2 G36-AS 5'AAAC taggtgggaggtgcggtgatC-3'
- 5/ NRE2 G42-S 5'-CACCG gcctgcgctcgttctggc-3'/NRE2 G42-AS 5'AAAC gccagaacggagcgcagggcC-3'
- 6/ NRE2 G46-S 5'-CACCG cccacctattgccccgcct-3'/NRE2 G46-AS 5'AAAC agggcggggcaataggtgggC-3'
- 7/ NRE2 G50-S 5'-CACCG cccacctattgccccgcct-3'/NRE2 G50-AS 5'AAAC agggcggggcaataggtgggC-3'

Plasmids were obtained from Addgene (PX458 and PX459 v2) and subcloning was performed as described at <https://www.addgene.org/static/cms/.../zhang-lab-general-cloning-protocol.docx>. The plasmids with confirmed sequences were transfected into angiomyolipoma and 293T cells. We used 293T cells to test cutting efficiency of the chosen guides because these cells can be easily transfected. Since these plasmids drive expression of GFP from chicken β-actin promoter, located upstream of Cas-9 promoter, we were able to confirm successful transfection of these plasmids into the cells using western blotting and immunofluorescence for detection of Cas-9 and GFP expression (Fig. II).

Figure. II Expression of Cas9 and GFP in (A) 293T and (B) 621-101 cells by western blotting. (C) Immunofluorescence detection of GFP in 621-101 cells.

Because our attempts involved the modification of endogenous promoter of Rheb, we also used an inhibitor of histone deacetylases to loosen DNA wrapped around histones and facilitate accesses of the guides to their targets, and therefore, enhance guides' cutting efficiency^(1, 2, 3). Cells were treated with 1uM of trichostatin twenty four hours post-transfection and assessed for cutting efficiency forty eight hours later. The DNA sequences of cells transfected with eleven combinations of seven guides (one combination per transfection), including highly transfectable 293T cells, revealed lack of efficient DNA cutting at the right position of the NRE2 region, which was indicated by lack of indel population (Fig. III).

Figure III: Assessment of NRE2 guide efficiency by the sequence trace decomposition. (A) The DNA in the cell pool consists of a mixture of indels, which yields a composite sequence trace after the break site almost similar with control. This indicates low efficiency indel generation under NRE2G4 CAS9 **(B)** Decomposition yielding the spectrum of indels and their frequencies; Inference of the base composition of +5 insertions -5 deletion. The percentage of 93.3% indicated low efficiency of indel generation. **(C)** Visualization of aberrant sequence signal in control (black) and NRE2-G4 sample (green), the expected break site (vertical dotted line) and the region used for decomposition (gray bar).

In summary, we were unable to examine directly the role of NRE2 in differentiation of angiomyolipoma cell using CRISPR approaches, perhaps, because of intrinsic difficulties related to targeting the particular NRE2 and NRE3 sites. For a different project, we have successfully applied CRISPER approach and this example is show in the rebuttal.

We used the Cas9 plasmid, containing several combinations of different guides, to facilitate cleavage of the human codon 72 *TP53* gene in 293T cells. The guides were designed using the tide website (<https://tide.nki.nl/>). The transfection of 293T and angiomyolipoma cells was performed as described above. The sequencing analysis of the targeted *TP53* region verified that out-of-seven guide constructs, three cleaved efficiently DNA at the

right codon 72 position. One example, the P53-g guide is shown in figure IV A and B. These three verified guides were used to facilitate arginine (R72) to proline (P72) change within codon 72 of TP53 in CRL4004 angiomyolipoma cells (Fig.IVD-E).

Figure IV. Efficiency of guides by sequence trace decomposition. (A) Due to imperfect repair after cutting by a targeted Cas9, the DNA in the cell pool consists of a mixture of indels, which yields a composite sequence trace after the break site shown after red arrow at p53-g-p53 compared to clean sequence of control p53-ctrl-p53; **(B)** Decomposition yielding the spectrum of indels and their frequencies; Inference of the base composition of +1 insertions -1 deletion. The percentage of 78.8% indicated without cutting the resting of percentage away from cutting site(+1 insertions -1 deletion indicate different indels population percentages).**(C)** Visualization of aberrant sequence signal in control (black) and P53-g sample (green), the expected break site (vertical dotted blue line) and the region used for decomposition (gray bar). **(D)** Screening strategy to identify CRL4004 clones with successful CGC to CCC substitution within codon 72 region. Green triangles indicate cells with arginine at the codon 72 position, red triangles indicate cells with proline at the codon 72 position, blue triangles indicate mixture of cells containing either R72 or P72. **(E)** Sequences of P72 (left), parental CRL4004 R72 (middle) and CRL4004 cells with CGC to CCC substitution (left panel). Double peaks (black: G; blue: C) indicate mixture of CRL4004 cells containing either R72 or P72.

References:

1. Nucleosomes Inhibit Cas9 Endonuclease Activity in vitro. *Biochemistry*. 2015 Dec 8;54(48):7063-6.
2. Nucleosomes impede Cas9 access to DNA in vivo and in vitro. *Elife*. 2016 Mar 17;5.
3. Nucleosome breathing and remodeling constrain CRISPR-Cas9 function. *Elife*. 2016 Apr 28

Because we failed to examine a direct role of NRE2/3 in angiomyolipoma differentiation through CRISPER, we used alternative Notch shRNA approach and overexpression of activated form of Rheb (Q64L). Angiomyolipoma cells depleted of Notch or expressing Q64L-Rheb were grown in N-medium, which promotes sphere formation, as described ⁽¹⁾. The depletion of Notch accelerated formation of non-adherent spherical clusters and neuronal differentiation of angiomyolipoma cells as determined by an increase in NS-tubulin-expressing cells compared to control (Fig. S4G).

Conversely, the overexpression of constitutively active form of Rheb (Q64L) impaired neuronal differentiation of CRL4004 angiomyolipoma cells in N-medium because the number of NS-tubulin-positive cells was reduced by 2-fold (Fig. S4H). Of note, the transient overexpression of Rheb does not allow for the extended follow-up of phenotypes, therefore, this experiment involved only the 24 hours exposure to N-medium after cell transfection (24 hours of transfection followed by 24 hours of the N-medium exposure). In addition, this experiment was performed only in CRL4004 cells because 621-101 cells have extremely low transfection efficiency while using Lipofectamine, 30% vs. 1-2%, respectively. Consequently, we could not express Q64L Rheb in 621-101 cells. Data from both experiments support functions of Notch and the Rheb-Notch-Rheb regulatory loop in controlling angiomyolipoma differentiation.

Reference:

1. Grandbarbe L, Bouissac J, Rand M, Hrabe de Angelis M, Artavanis-Tsakonas S, Mohier E. Delta-Notch signaling controls the generation of neurons/glia from neural stem cells in a stepwise process. *Development*. 2003;130(7):1391-402.

3. The authors stated that deletion of TSC1 in nestin expressing podocytes resulted in the formation of cysts, RIN, and renal cancer. However, the data in Fig. 7 show that the lesions are likely to be originated from tubular cells, not podocytes. There are many intact glomeruli surrounding the lesions. In addition, renal tubular cell-specific TSC1 knockout but not podocyte-specific TSC1 knockout causes the aberrant cystic formation. The authors should demonstrate TSC1 ablation in podocytes in nestin-cre mice by staining p-S6 or carefully discuss the origin of the cysts and tumors in these mice.

We agree with the reviewer and have revised the sections of the manuscript pertaining to the origin of RCC in our mouse model. The nestin expression is not limited to podocytes. The renal progenitors localized in the Bowman's capsule urinary pole also express nestin. These progenitors can regenerate both tubular cells and nestin-positive podocytes (reference # 52), therefore it is likely that we induced the formation of lesions that morphologically resemble tubular-origin cysts or cystadenomas by targeting these progenitors. The origin of renal lesions from nestin expressing progenitors is supported by the single nestin positive cells, which have

spindle-shape morphology, similar to these progenitors, within the malignant renal lesions (Fig. 7D-G, nestin [red], pS6 [green]).

Furthermore, in contrast to the previous studies the renal lesions observed in our model have clear histological features of malignancy, which were not present in the lesions described previously, underscoring different biology, and likely different origin of the tumors in our model. We also determined that these lesions express cre recombinase (Fig. 7H), which undermines probability of off-targets effects.

Reviewer #2 (Remarks to the Author):

TSC-RHEB-NOTCH ms rev

0. Since there is no evidence that any of the data generated is of a normal distribution, and consistent with full presentation of all data, please add/replace all graphs that show mean and SE marks with dot plots showing all data points. E.g. 1Dii, 1F, 1G, 2A,

All graphs have corresponding dot plot graphs. We either replaced graphs in the main figures with dot plots, or provided the additional dot plot graphs (Fig. S7)

1. Figure 1Dii. the percentage of co-expression seems quite low, 0.2-6%. Is this significantly above background for two randomly chosen proteins assess by IHC?

The percentage of co-expression of nestin and peripherin may appear to be low. However, the data represent the AREA occupied by the double-stained cell, which correlates with the percentage of cells that co-express both markers, however, does not reflect the actual percentage. This analysis can be easily performed and was done in a blind fashion using automatic unbiased approach (Aperio digital pathology). This methodology is FDA approved for the evaluation of estrogen/progesterone receptor and Her2 expression in breast cancer patients' samples. As expected, nestin positive cells were rare, as depicted by the strong brown coloration (Fig. 1Di). Conversely, the majority of cells expressed peripherin. For the purpose of this analysis, only strongly positive cells were considered, therefore, the potential noise/background interference has been

eliminated. The background noise is routinely removed prior to analysis, therefore the area of co-localization is free of background error. Furthermore, the Aperio digital pathology system uses the algorithm based software analysis to locate and quantitate staining. We used the co-localization algorithm, which separates the stains based on the RGB values into three channels, corresponding to the colors used, allowing us to quantify the area and intensities of each stain separately and together. After running the algorithm analysis, the software generates a 'Markup image' with an arbitrary color for each stain and for areas where stains co-localize. The negative areas with only hematoxylin staining are marked in "Blue", "Red" and "Green" for individual stained areas and "Yellow" in the areas where the stains superimpose. We added the appropriate description to our material and methods.

I don't see any yellow in 1Di 'Markup' only some green. How does red + brown give either yellow or green?

This analysis was performed using the co-localization algorithm, which separates the stains based on the RGB values into three channels corresponding to the different colors. The colors assigned by the software and shown in Markup image are so called pseudo-colors that do not reflect the colorations resulting from the enzymatic reaction within a section during an immunostaining procedure (those observed by eye under light microscope). We incorporated high power images for easier identification of overlapping stainings (yellow) and marked these areas with arrows (Fig.1D-i, right panel).

Please see also our response to comment 1 of the reviewer 2.

2. Figure 2A legend states: Data are representative of three, n=3 (A-iii-vi, C, D). Please average together all of the data points and show in the figure rather than just a representative set.

The data shown in the figure 2A represents all experiments that we have done, therefore this is not a representative set. The appropriate revision is included in the figure description.

Are the differences at the different times statistically significant?

We tested the statistical significance of differences in the expression of *RHEB* and *HES1* at different time points and found that the expression of *RHEB* was significantly higher at 30, 120, 150, 180, 210 and 240 minutes after release from serum synchronization compared to the time point zero (Fig. 2A-i, iii). Similar to *RHEB*, the expression of *HES1* was significantly higher at 30, 90, 120 and 180 minutes after release from serum synchronization (Fig. 2A-ii, iv). The oscillation of *Rheb* and *Hes1* was confirmed by western blotting (Fig. 2B). We also tested the statistical significance of differences in the expression of *RHEB* and *HES1* in DMEM vs. in N-medium and found that the expression of *RHEB* and *HES1* was significantly higher at 30, 90, 120, 150, 180, 210 and 240 minutes after release from serum synchronization in DMEM (Fig. 4A and, S5A and B).

Do GAPDH levels change over time? This could explain differences in RHEB and Hes1.

The GAPDH expression did not change over the time (Fig. S5C-D)

3. What is the number of NREs in the human genome, and the distribution of such elements seen within 5 kb of the TSS for all human genes?

Dr. Brandon White, who collaborates with us on this work, and is listed as an co-corresponding author on this manuscript, performs a separate study, which aims to identify the number of NREs in the human genome and the distribution of these elements within 5 kb of the TSS for all human genes. The results of his study will be published in the separate manuscript that originates from his laboratory. Based on the personal communication with Dr. White we are aware of the nine NREs, six in forward (Head) and three in reverse (Tail) direction, within 5kb of TSS of RHEB (upstream of TSS).

4. a/ The binding shown in Figure 2Bii seems quite modest. Nonexistent for NRE1. Please perform this experiment for other validated NOTCH binding sequences in other genes, and compare with the enrichment seen here.

The binding of Notch to the Hes1 vs. RHEB promoter in non-synchronized cells was compared. This additional data has been incorporated into the figure 2C-ii-iii. We also added the new panels illustrating the binding of Notch1 to the promoters of RHEB and Hes1 in CRL4004 cells (**Fig. 2C-iii-v**). Please see also our response to the comment 1 of the reviewer 1. We discussed there how the presence or absence of paired sites within the promoters affects the magnitude of Notch-dependent binding to these promoters and activation.

b/ How can NRE1 ablation as in Ci have an effect on RHEB expression when it does not bind NOTCH, as in Bii? (Former 2C-i and 2B-ii)

The reviewer 3 (comment #2) was concerned about performing the luciferase assay only in 293T cells, in which there is a chance of possible regulation of RBPJ-dependent genes in the Notch independent fashion. The second concern regarding this cell line referred to the commonly observed very high expression of transfected genes when plasmids with the large T origins of replication are used. We acknowledge issues raised by the reviewer, therefore, we performed experiments using heterologous cell line HeLa cells and a new system for the reporter assays, developed in Dr. White laboratory, to confirm our conclusions. We replaced all luciferase reporter assay data from 293T cells with data from HeLa cells. New data indicate that the activity of NRE1 mutant is similar to the wild type RHEB promoter, suggesting lack of importance of NRE1 site for regulating RHEB promoter activity (**Fig. 2D-i**). Please see also the **comment 1 of reviewer 1**.

These new results are consistent with data included in figure 2C-ii, iv (please see comment 4a of the reviewer 2) and figure 2E-i that show weak binding of the endogenous Notch1 to the endogenous NRE1 site in angiomyolipoma cells.

5. The methods do not provide sufficient details on the constructs and methods shown in Figure 2Ci and ii.

We have included detailed information regarding the human Notch 1 intracellular domain and Mastermind-like 1 constructs. The catalog numbers for the luciferase RHEB and control promoter constructs are included into the materials and method section.

6. Why is the expression of RHEB reduced in the N1ICD+MAML1 expression experiment, c/w N1ICD alone in figure 2Ci, whereas Ci shows enhanced activity of the promoter with both expressed?

This panel was replaced with new data (**Fig. 2D**) due to concerns of using 293T cells for the reporter assays. Data included in this figure consistently indicate increased or decreased RHEB expression upon overexpression or downregulation of Notch1, respectively. Please see also our response to the comment 4b of the reviewer 2.

7. For Figure 2D, please show a control to demonstrate effects of DAPT on a canonical NOTCH-NRE promoter.

The former figure 2C-D was replaced with the new figure 2D (please see response to comment 6 of the reviewer 2). Please see also our response to the comment 4b and 6 of the reviewer 2. To eliminate off-target effect of DAPT and to confirm Notch-dependent regulation of Rheb expression, we downregulated Notch using shRNA in several cell lines (Fig. 2D-iii-iv). Downregulation of Notch reduced the expression of Rheb and canonical Notch-NRE target, Hes1 (Fig2. D-iii-iv).

8. Figure 2Ei, the 210 data point for NRE1 does not look statistically significant. Again, dot plots should be shown for this data, and would be very informative.

We found the significant differences in binding of Notch 1 to the *RHEB* and *HES1* NREs in DMEM- vs. N-medium (Fig. 4B, $P=0.0462$ for NRE3 and $P=0.0023$ for NRE2, by two-way ANOVA).

Binding to the NRE1 was weak and did not increase or oscillate (Fig. 4Bii), therefore, we concluded that NRE1 does not play an important role in regulating Rheb, in contrast to NRE2 and NRE3. The manuscript has been revised accordingly to reflect our conclusions and to incorporate new data. The dot plots for all panels are shown in supplementary figure S7C.

9. The 621 cells are well-known and well-characterized. The CRL4004 and CRL4008 cell lines, on the other hand, are not. Please provide documentation that these cell lines are derived from angiomyolipoma: 1) show biallelic mutation/inactivation of *TSC2*, 2) show by immunoblotting that they do not express *TSC2*, and have unregulated activation of mTORC1.

The following angiomyolipoma-derived cells were used in the original manuscript: (1) 621-101 with the inactivation of both *TSC2* alleles (Yu et. al., 2004), (2) CRL4004 (ATCC) with a 5bp deletion (4083-4087) in the

exon 33 of *TSC2*, likely affecting the tuberlin GAP domain (Lim et. al., 2007), with the haplo-insufficiency effects, derived from *TSC*-associated angiomyolipoma, and. (3) CRL4008 (ATCC) from a sporadic angiomyolipoma (Arbiser et. al., 2001).

We performed comprehensive *TSC1/TSC2* genetic analysis, in collaboration with Dr. David Kwiatkowski, by massively parallel sequencing (MPS), using a customized gene bait set, yielding > 500x coverage of the entire genomic extent of *TSC1* and *TSC2*. This analysis confirmed the known frameshift *TSC2* mutation c.4081_4085delCGAGT/ p.V1362Lfs* with allele frequency 48% (hg19), previously reported by Lim et al.2007, in the CRL4004 cell line. We did not identify small/point mutations or large deletions in the CRL4008 cell line. The allele frequency SNP analysis of *TSC1/TSC2* genes showed no hint for LOH in either cell line (CRL4004 and CRL4008) (skewed allele frequency AF<0.4 or AF>0.6). This highly sensitive analysis (Tyburczy et al. 2015) strongly suggests that the CRL4004 cell line has one inactivated and one normal allele of *TSC2*. The CRL4008 cell line does not appear to have any mutations in the *TSC* genes.

Therefore, we focused on the two angiomyolipoma-derived cell lines: 621-101 and CRL4004. CRL4004 cell line appears to be suitable angiomyolipoma model because it shares multiple common features with the 621-101 cells such as reduced expression of tuberlin in comparison to control 293T and HeLa cells (Fig. S1E) and increased mTORC1 activation in comparison to normal human fibroblasts. Serum deprivation did not reduce phospho-S6K in the CRL4004 cells (Fig. S1F). In addition, the IPA (pathway analysis) of CRL4004 RNA identified hyperactivation of mTOR, EIF2, p70S6K and PI3K/AKT signaling. The activation of these pathways in CRL4004 was similar to 621-101 and sporadic angiomyolipoma compared to corresponding normal kidney from the same patient (Table S2). The expression of some of these genes was confirmed by RT-qPCR analysis of CRL 4004 and 621-101 cells, sporadic angiomyolipoma, and corresponding normal kidney from the same patient (Fig. S1G). Finally, western blotting analysis confirmed upregulation of Rheb and Raptor in CRL 4004, 621-101, and sporadic angiomyolipoma compared to corresponding normal kidney from the same patient (Fig. S1H).

10. Why do NS-Tubulin levels rise to ~40% on day 7 in 3Cii, yet only ~7% in Dii?

In the original figure 3D (Fig. S4F) cells were stably transfected with control or Rheb shRNA and underwent selection process, which may lead to the selection of the population with the specific differentiation preferences. This population may differ from the parental cells, used in experiments shown in original figure 3C. Since this experiment takes at least seven days to complete, we are not able to perform this study without pressure of selection and under transient downregulation of Rheb. In addition to experiments involving depletion of Rheb, overexpression of Rheb was induced (please see our response to comment 12 of the reviewer 2) and differentiation profile of cells, overexpressing activated Rheb (Q64L) in N-medium, was examined (Fig. S4H).

11. The lack of RHEB and Hes1 cycling in Figure 4Ai is expected since the cells are not going through a serum-deprive-addback treatment, and can't expect this cycling to occur.

The cells grown in N-medium were deprived of chicken embryo extract for twenty four hours, to mimic serum-deprivation of angiomyolipoma, grown in DMEM medium, followed by addition of chicken embryo extract. To address the reviewer's concern, we repeated these experiments using angiomyolipoma cells, grown in N-medium for seven days, followed by twenty four hours chicken embryo extract deprivation, to synchronize cells, and subsequent fetal bovine serum stimulation to release cells from synchronization. Similar to chicken embryo extract, fetal bovine serum did not induce *RHEB* and *HES1* mRNAs oscillations in angiomyolipoma cells grown in N-medium. The suppression of *Hes1* oscillation, and, therefore, Notch signaling, is required for adoption of neuronal fates by neural stem cells (Shimojo *et al.*, 2008). Thus suppression of *Hes1* oscillation in angiomyolipoma cells in N-medium is consistent with these published findings. Additional data are included in the figure 4A (green lines). The manuscript has been revised accordingly to improve clarity.

12. The observations in Figure 4, and corresponding text are interesting but observational only, and it is likely that many changes in many signaling pathways occur with the switch to N-medium. To connect this to the NOTCH-RHEB pathway directly, does overexpression of RHEB block the phenotypes seen in N-medium?

We performed the suggested experiment. The differentiation profile of cells overexpressing activated form of Rheb (Q64L) in N-medium was examined. We choose CRL4004 over 621-101 cells because of much higher transfection efficiency of CRL4004 cells compared to 621-101 cells while using Lipofetamine, 30% vs. 1-2%, respectively. We found that the constitutively active Rheb (Q64L) blocked neuronal differentiation of CRL4004 angiomyolipoma cells grown in N-media, as the number of NS-tubulin-positive cells was reduced by 2-fold (Fig. S4H).

The transient overexpression of Rheb does not allow for prolonged follow up of phenotypes. Therefore, this experiment involved only 24 hours exposure to N-medium after cell transfection (24 hours of transfection followed by 24 hours of N-medium exposure). These data are consistent with accelerated neuronal differentiation of angiomyolipoma cells resulting from Rheb depletion (Fig. S4F).

13. Figure 1ABCD. It would be nice to have quantitative information on the expression of all of these neural-related antigens than just IHC. Please add figures showing expression of them all by immunoblotting, similar to what is seen in Cii.

The expression of GFAP and neuron specific enolase in angiomyolipoma tumors by Western blotting was shown in the initial submission and is included in the revised manuscript (Fig. 1 C-i-ii). We performed additional immunoblotting analysis on three angiomyolipoma tumors that were available to us to verify the expression of nestin. This analysis confirmed that angiomyolipoma tumors express higher levels of nestin compared to the corresponding normal kidney (Fig. 1C-ii and E-ii, new panel).

The IHC analysis of NS-tubulin expression revealed that in non-diseased kidney only peripheral nerves express NS-tubulin, indicating high specificity of this antibody ((Fig. S1A, arrow). We also detected the expression of NS-tubulin in small groups of cells within angiomyolipoma tumors (Fig. S1A, arrow), including cells located within neoplastic vessels (Fig. S1A, arrowhead). In addition, we found micro-papillary lesion expressing NS-tubulin within the surrounding kidney tissue (Fig. S1A, arrowheads).

We also verified using western blotting that both angiomyolipoma cell lines express Nestin, GFAP and NS-tubulin (Fig. 1E-ii). We were unable to verify the expression of these markers by western blotting in LAM samples because we do not have access to fresh LAM tissue.

14. Note that many previous studies have shown differentiation abnormalities in neural precursor cells lacking *Tsc1/Tsc2*. Mol Cell Neurosci. 2002 Dec;21(4):561-74. is one of the earliest.

This reference is discussed in the revised manuscript.

15. What is the control in Figure 6Aiii? It looks very different from the control in 6Ai. There is a comment about strain background effects, but that is not credible for the difference.

The corresponding Cre-negative littermate controls for each strain were used in both, 6A-i and 6A-iii (Cre-negative embryos were compared to Cre-positive embryos from the same litter). Due to different pace of *Hes1* and *Rheb* mRNAs oscillation in each individual embryo, the expression of *Rheb* and *Hes1* are now shown separately for each individual animal for better clarity and transparency (Fig. 6A and B). The statistical analysis of repeated measures (longitudinal data) demonstrated that *Hes1* oscillation patterns between controls and mutant mice are significantly different ($P=0.0146$). In addition, the regression analysis revealed statistically significant correlations between *Rheb* and *Hes1* in both control ($R^2=0.579$, $P<0.0001$) and mutant mice ($R^2=0.36$, $P=0.02$). The mean expression of averaged *Rheb* and *Hes1* was significantly different in *Nestin-Cre;Tsc1^{fl/fl}* embryos vs. wild type littermate controls ($P=0.0201$). These data indicate that changes in *Rheb* and *Hes1* expression occur parallelly and that oscillation patterns are influenced by *Tsc1* loss. The increase in *Hes1* and *Rheb* expression in *Tsc1*-null cells was reduced as a result of Notch1 haploinsufficiency (in *Nestin-Cre;Tsc1^{fl/fl};Notch1^{fl/+}* embryos) and was similar or lower to the expression of *Rheb* in wild type littermate controls at all the time points (Fig. 6B and 6C-ii).

To support the role of *Rheb* oscillation in preventing *Tsc*-null cells differentiation, we performed western blot analysis of *Rheb* and *Hes1* proteins in synchronized neural tube-derived cells from littermate control and *Tsc1^{-/-}* mice and confirmed the oscillatory pattern of expression in *Tsc*-null cells (Fig. 6C).

Using densitometry analysis we confirmed that expression of *Rheb* at 120 minutes was increased by 5-fold in *Tsc1*-null cells vs. 2.8-fold increase in littermate control cells relative to 0 minutes (Fig. 6C-i).

15a. Please show data for Hes1 in the Nestin-cre Tsc1ff Notch1+ cells.

We added the *Hes1* mRNA and Hes1 protein oscillation to (Fig. 6B-iii-iv and 6C-ii). Notch1 haploinsufficiency reduces Hes1 expression by 2-fold at 90 minutes (Fig. 6-iv).

16. Please show immunoblot data for pS6, Rheb, Tsc1, Tsc2, mTORC1 signaling for the embryo cell cultures shown in Figure 6.

We included immunoblot data for hamartin, tuberin, pS6 kinase, Rheb, Notch1 and Hes1 from mouse embryo culture cells (Fig.S6E)

17. Show blot data for all the proteins analyzed in 6C on the ELT3 cells to confirm that there is an increase in expression.

We included Nestin, MelanA and GFAP immunoblots from ELT3 cells (Fig. 6E-ii). We used mouse embryo neural tube and brain as a positive control. Nestin is highly expressed in ELT3 cells, which is consistent with data obtained by FACS. We did not include smooth muscle actin, as its expression is well established in ELT3 cells

(Howe, S. R., Gottardis, M. M., Everitt, J. I., Goldsworthy, T. L., Wolf, D. C., and Walker, C. (1995) *Am. J. Pathol.* **146**, 1568-1579).

18. The authors ignore J Clin Invest. 2010 Jan;120(1):103-14. doi: 10.1172/JCI37964, which also showed a role for Notch in TSC signaling, and a response to DAPT treatment.

This reference is included in the revised manuscript.

19. Figure 7 does not indicate the age of the mice at the time of injection with tamoxifen, or the age of the mice at the time of sacrifice.

Tamoxifen was injected intraperitoneally into 2-3 month old mice at the dose of 120mg/kg/day for two consecutive days. Mice were sacrificed at the age of seven months. This information is now included in the legend of the figure 7C.

20. What was the survival of all of the brain model mice? Was this significantly improved by presence of the Notch1f allele?

The *Nestin-Cre;Tsc1^{ff}* mice did not present with any central nervous system manifestations nor decreased survival. We did not tested if the partial loss of *Notch1* function improves phenotypes induced by loss of *Tsc1*.

21. The mouse kidney model data (Figure 7) could have occurred due to off-target expression of cre in any kidney epithelial cell. The pathology is similar/identical to that seen in multiple kidney models over the past 15 years including the *Tsc2^{+/-}* and *Tsc1^{+/-}* mice.

This is addressed in the response to the comment #3 of the reviewer 1.

Reviewer #3 (Remarks to the Author):

Cho et al. propose the existence of a Rheb-Notch-Rheb autoregulatory loop that regulates the differentiation of neural crest derived progenitors and is instrumental in generating phenotypes associated with tuberous sclerosis. These studies follow up on prior work from the PI suggesting the TSC/Rheb act upstream of Notch in Tuberous sclerosis and in *Drosophila*. This is an intriguing hypothesis, but much of the data in the current manuscript as it relates to Notch is problematic.

1. Figure 1 selects a handful of markers, some not specific such as NSE, to back up the idea that AML cell lines are plastic with respect to differentiation. What does you see if you use an unbiased approach (e.g., RNAseq or expression profiling)?

To support differentiation plasticity of angiomyolipoma, we performed a global transcriptomic analysis of sporadic angiomyolipoma, corresponding normal kidney, and two angiomyolipoma-derived cell lines, using RNA sequencing (RNA-seq). The new supplementary figure 2 depicts a heatmap illustrating genes differentially

expressed in angiomyolipoma or angiomyolipoma-derived cells vs. normal kidney. Three gene lists were intersected and only common genes, which were differentially expressed in angiomyolipoma and angiomyolipoma cells relative to normal kidney (~1400 genes), were used for the gene ontology (GO) enrichment analysis. The analysis of the significant genes was performed using GOSTATS package (Bioconductor software). The enriched biological processes were defined by P cutoff of <0.001 and only the enriched categories related to neurogenesis, multipotency and differentiation were shown (Fig. S3). These processes included axon extension, regulation of neuron and cell differentiation, morphogenesis and nervous system development (Fig.S3).

Using pathway analysis, we found also alterations in expression of genes involved in stem cell pluripotency, melanocyte, pigmentation, and renal cell carcinoma signaling (Table S2). In addition, the bio-function categories enriched for differentially expressed genes in angiomyolipoma and angiomyolipoma cell lines are shown (Table S3). In conclusion, these unbiased analyses of global gene expression support differentiation plasticity of angiomyolipoma.

2. The reporter experiments performed in 293T cells are problematic for several reasons. These cells express viral E1A 13S and E1A 12S proteins that bind and variously activate or inhibit RBPJ-dependent genes in a completely Notch independent fashion, confounding clear interpretation of the results of Notch reporter gene assays (Ansieau et al. Genes & Development 15:380, 2001). 293T cells also drive very high expression of transfected genes if plasmids with large T origins of replication are used, raising questions about the relevance of the observed modest activation of the Rheb promoter element in the experimental data provided. Additional experiments in another heterologous cell type are needed to confirm conclusions drawn from the 293 cell experiments.

Data included in the original figure 2C-i-ii were replaced with data from experiments in HeLa cells (Fig. 2D-i) and several other heterologous cell lines (Fig. 2D-ii-iv). These new data support the conclusion that NRE2 and NRE3 sites are responsible for Notch-dependent transactivation of Rheb while NRE1 is not. We confirmed Notch-dependent regulation of Rheb expression using Notch shRNA approach in several cell lines (Fig. 2D-iii-iv).

Downregulation of Notch reduced the expression of Notch1, Hes1 and Rheb (**Fig2D-iii-iv**). Please see also our response to the comment 1 of the reviewer 1 and 4b of the reviewer 2.

3. The idea that NRE1 binds Notch1 and is inhibitory is inconsistent with all past data regarding Notch transcription complexes and transcription; there are no well-characterized examples of Notch complexes having repressive effects. Moreover, to truly prove that NRE1 and NRE2 are involved in Rheb regulation in the appropriate cellular context would require other approaches, such as CRISPR mutagenesis putative Notch binding elements in AML lines.

We agree with the reviewer. There is no evidence to suggest that the Notch ternary complex can be inhibitory via NREs. However, in the absence of Notch, the CSL/RBPJ protein can act as an inhibitor by recruiting inhibitory complexes. Our new data (**Fig. 2D**) indicates that the activity of NRE1 mutant is similar to wild type Rheb (please see also our response to the comment 1 of Reviewer 1 and 4a-b of Reviewer 2). Furthermore, our data indicate very weak binding to the NRE1 (**Fig.2C-ii, 2E-i**), therefore, we did not pursue our study into this direction. Instead, we re-focused our manuscript on NRE2 and NRE3. The manuscript has been revised accordingly. Regarding CRISPR please see our response to the comment 2 of Reviewer 1.

4. Nowhere do the authors actually measure activated Notch1, since they rely on a Santa Cruz antibody reagent that recognizes the intracellular C-terminus of Notch1. Many of the western blot analyses need to be redone with a NICD1 specific antibody, such as that available from Cell Signaling Technologies.

The conclusion about cleavage and activation of Notch1 for the initial submission were made based on the size of the detected Notch1 and the expression of Hes1. The size of Notch1 ICD detected using Santa Cruz antibody was around 110 kDA, which corresponds to activated Notch1 receptor and is different from the size of full length receptor (~300 kDA). However, as suggested, we used antibody that specifically recognizes cleaved Notch1 receptor at the Valine 1744 residue (Cell Signaling Technology, catalog #4147). The results obtained using this antibody (**Fig. 4D and G**) correspond well to our results obtained using Santa Cruz antibody.

5. Is Rheb really oscillating? There are some modest inconsistencies in the upward trend of Rheb expression after stimulation in DMEM, but there are no nadirs akin to what is seen with Hes1, which truly does oscillate,

except in one experiment (Fig. 6Aiii, current 7A-iii).

The oscillation of Hes1 mRNA is more evident than Rheb as a result of the higher amplitude of Hes1 oscillation (Fig. 6A and B). We acknowledge that low amplitude of Rheb oscillation may hamper data interpretation. However, the regression analysis established that Hes1 and Rheb change parallelly, therefore, we conclude that Rheb oscillates similar to Hes1 (see the response to the comment 15 of the reviewer 2). We confirmed the oscillation of Rheb and Hes1 by western immunoblotting (Fig. 2B and 6C). We found that both, Rheb and Hes1, have oscillatory pattern of expression in angiomyolipoma-derived cells (Fig. 2B), and in *Tsc1*-null mouse embryo progenitors (Fig. 6C). This conclusion is also supported by the lack of Rheb and Hes1 oscillation (the peaks of mRNA expression) in N-medium (Fig. 4A) or suppression of Hes1 and Rheb oscillation in the *Tsc1*-null mouse embryo progenitors with partial loss of Notch1 function (Fig. 6B and 6C-ii). Please see also the response to the comment 2 and 15 of the reviewer 2.

6. The authors use qChIP-PCR to show loading of Notch1 onto NRE2 and not NRE1 at most time points following cell stimulation (Fig. 4Bi and Bii). These two sites are only 212 bps apart. What is the size of the chromatin-DNA fragments used in these experiments? It might normally be difficult to distinguish between loading of sites that are this close together.

We agree with these concerns. However, we observe the consistent and stronger binding of Notch1 to NRE2 compared to NRE1 on the endogenous promoter Rheb. This binding is reduced in N-medium. The DNA gel image indicates that the size of digested/sonicated chromatin that was subjected for our ChIP-qPCR analyses is between 150-200 bp.

7. With respect to changes in Notch as they relate to neuronal differentiation (Figure 4D), the data suggest that Notch1 expression is downregulated as the cells adopt a neuronal fate. Notch signaling in other contexts has been reported to be important in upregulating mTOR and S6 phosphorylation as well as cell growth. Hence, a simpler explanation for the observed changes in phenotype is that the N-medium downregulates Notch1 expression through currently unknown mechanisms, leading to downstream effects that include loss of mTOR activity. Rheb knockdown may convey a similar phenotype by also interfering with mTOR and its downstream effects. Stated another way, the more complex Rheb-Notch-Rheb signaling axis proposed by the authors and the need for oscillatory loading of Notch1 onto Rheb may be an unnecessary complication.

We agree with the reviewer that it remains unclear how N-medium favors neuronal differentiation. However, the oscillation of Notch have been implicated in maintaining neuronal cells in undifferentiated (stem cell) state (*Shimojo et al.*, 2008), therefore, we propose that the similar mechanisms operate to maintain undifferentiated state of angiomyolipoma cells, and, consequently, that the reciprocal interactions between Notch and Rheb contribute to these mechanisms.

We also agree with the reviewer that downregulation of Rheb will decrease mTOR signaling, and therefore, it is impossible to entirely rule out a role of mTOR in angiomyolipoma cell differentiation. Although we recognize that Rapamycin is not ideal tool to block mTOR signaling, our data included in supplementary figure 4I suggest that blocking mTOR with rapamycin does not accelerate differentiation of angiomyolipoma cells in N-medium in contrast to downregulating or overexpressing Rheb (Fig. S4F and S4H).

8. The spontaneous renal tumorigenesis shown in figure 7 is impressive, but does not address the proposed model. It would be of interest to know if, for example, conditional haploinsufficiency for Notch1 suppressed or modified tumorigenesis in the mouse.

The mouse model included in the manuscript underscores the importance of targeting *Tsc1* in cell specific manner to induce tumorigenesis and provides some initial evidence regarding the origin of renal lesions in TSC. We absolutely agree with the reviewer that the inclusion of data demonstrating that the conditional

haploinsufficiency for Notch1 suppresses or modifies tumorigenesis in this mouse model would strengthen this work. However, we feel that the amount of work required to perform this task preclude conducting these studies in a reasonable time frame. We would like to point to the reviewer and editors that the similar *in vivo* approach (conditional Notch1 haploinsufficiency) has been used in experiments shown in the figure 6B-C. Please see also our response to the comment 20 of the reviewer 2.

Reviewers' comments:

Reviewer #1 (Remarks to the Author):

The authors made large efforts to rigorously respond to the questions raised by the reviewers, and I believe that the paper has been strengthened.

However, I have still a few questions regarding with the new data shown in Fig. 7.

1. In Fig. 7 D, E, and F, why do the majority of netin-positive cells show little mTORC1 activation? Co-localization of netin and pS6 shown in Fig. 7G is not convincing. Are these data consistent with the data and conclusions derived from Fig. 6?

2. The results of Fig. 7H is ambiguously described. In the discussion, the authors stated "our data suggest the potential origin for TSC-associated RCC from renal progenitors localized in the Bowman's capsule urinary pole".

In Fig. 7H, I recognize four intact glomeruli (two are in left bottom and the other two are in upper right) in the image, and one glomerulus located at the upper right seems to have a part of the urinary pole, but little nestin in that area. The parietal epithelial cells in the Bowman's capsule express very low levels of nestin while the proximal tubules (may include some distal ones) express high levels of nestin. It would be helpful if the figure includes some terminologies and indications for the nestin-expressing organelles as well as the region of tumor.

This reviewer feels that the current data in Fig. 7 do not clearly support and indicate the role of the Rheb-Notch-Rheb loop amplification in the onset /development of TSC-renal tumors.

Reviewer 1 comments on responses to Reviewer 2 comments:

The reviewer commented for the editor only and was satisfied with the authors responses.

Reviewer #3 (Remarks to the Author):

Cho et al. have been responsive to the initial set of criticisms and the paper is strengthened as a result. A number of issues remain.

1. Without genetic evidence, e.g., using CRISPR mutagenesis of the endogenous RHEB promoter, it remains unproven that Notch regulates RHEB via NRE2 and NRE3 (though the data presented are consistent with the possibility that these elements may regulate RHEB, at least to some degree). Parenthetically, it would be of interest to know if double mutation of NRE2 and NRE3 results in reduction of RHEB promoter activity below the ~50% reduction seen with single mutations in these sites.

2. There is additional confusion caused by other nomenclature used by the authors. In Figure 2Dii and 2Dii, it appears that the label N1ICD refers to the transfected Myc-tagged activated Notch1 protein, whereas in Figure 2Diii N1ICD presumably refers to endogenous Notch polypeptides in each cell type that are recognized by an antibody against an epitope located in the intracellular region of Notch1. Most of these polypeptides do not correspond to activated Notch1 (they are mostly the transmembrane subunit of furin-cleaved full-length Notch1) and some other label should be used. By contrast, the V1744 antibody used in Figure 4D and 4G is specific for N1ICD, and if this name is to be applied to polypeptides in any of the Western blots presented, it should be applied here. It also is confusing to call any sequence that resembles an RBPJ binding site an NRE; this name should be reserved for sites that are proven to be functionally important.

3. In Figure 2D, it appears that the effects of shRNA knockdown of Notch1 on Rheb is quite a bit more dramatic at the protein level (Figure 2Diii) than it is at the RNA level (Figure 2Div). The description of the data doesn't acknowledge this discrepancy, which implies that the interaction

between Notch and Rheb protein levels may involve mechanisms beyond transcription regulation of Rheb by Notch. In a similar vein, in Figure 4G, it is evident that N1ICD (activated Notch1) is sharply downregulated on day 1, yet there appears to be no change in Rheb protein levels at this time point. How long does it take for N1ICD levels to fall? Is Rheb a stable protein (and therefore the fall in Rheb lags well behind the drop in N1ICD)?

Other minor comments

In Figure 4I, the microarray data are redundant with 4J and can be deleted.

Figure S3 is mislabeled.

The source of the cleaved Notch1 antibody is not given in the methods

We thank the reviewers for positive comments and constructive criticism. Please find our responses to reviewers' comments below in blue.

Reviewer #1 (Remarks to the Author):

The authors made large efforts to rigorously respond to the questions raised by the reviewers, and I believe that the paper has been strengthened.

However, I have still a few questions regarding with the new data shown in Fig. 7.

1. In Fig. 7 D, E, and F, why do the majority of netin-positive cells show little mTORC1 activation? Co-localization of netin and pS6 shown in Fig. 7G is not convincing. Are these data consistent with the data and conclusions derived from Fig. 6?

We incorporated into the revised manuscript several high power images, acquired using confocal microscopy, representing single layer from a Z-stack, for easier identification of mTORC1 activation within nestin positive cells (**Supplementary Figure 8**). The arrows in each panel point to nestin-positive cells that also contain phospho-S6. The images showing single fluoresce (green or red) along with the merge images that show nuclei (DAPI) indicate that both proteins are present in the same cells, although only in the lowest panel clear co-localization (yellow, arrowhead) is observed in the merge image. Nestin positive cells are relatively rare within tumor parenchyma, as they may represent undifferentiated stem cells that tend to be rare within better differentiated tumors.

The data in Figure 7 point to the possible origin of renal lesions from nestin expressing progenitors and are not intended to address the role of the Rheb-Notch-Rheb regulatory loop in TSC tumorigenesis in

contrast to the figure 6. The manuscript was modified accordingly for clarity and to include new data (the description of the results included in figure 7 was moved to the separate subsection).

2. The results of Fig. 7H is ambiguously described. In the discussion, the authors stated “our data suggest the potential origin for TSC-associated RCC from renal progenitors localized in the Bowman’s capsule urinary pole”.

In Fig. 7H, I recognize four intact glomeruli (two are in left bottom and the other two are in upper right) in the image, and one glomerulus located at the upper right seems to have a part of the urinary pole, but little nestin in that area. The parietal epithelial cells in the Bowman’s capsule express very low levels of nestin while the proximal tubules (may include some distal ones) express high levels of nestin. It would be helpful if the figure includes some terminologies and indications for the nestin-expressing organelles as well as the region of tumor.

As discussed in our first rebuttal letter (a response to the comment 3 of the reviewer 1), data in the figure 7H show the expression of Cre recombinase (not nestin) within the renal lesions of Tsc1-null mice to document that tumor cells origin from Cre-expressing cells and exclude possibility of off-targets effects of Cre recombinase. As reviewer noticed, Cre recombinase is expressed at high levels within dysplastic tubules adjacent to papillary neoplastic lesion indicating that these abnormal tubules origin from Cre expressing progenitors. The less evident expression of Cre in intact glomeruli is expected, as podocytes and the urinary pole progenitors have smaller cytoplasm comparing to large tumor cells and tubules. Although the cells that give rise to tumors in Tsc1-null mice are expected to express nestin, only rare tumor cells preserved this expression, likely because of inactivity of the endogenous (mouse) nestin promoter upon differentiation of tumor cells in mice. Of note, Cre expression in this mouse is driven by transgenic rat promoter, therefore, this expression may be differently regulated from endogenous nestin expression, detected by immunofluorescence. We revised manuscript accordingly for clarity.

This reviewer feels that the current data in Fig. 7 do not clearly support and indicate the role of the Rheb-Notch-Rheb loop amplification in the onset /development of TSC-renal tumors.

We agree with the reviewer and would like to emphasize that the purpose of the figure 7, as discussed in the response to the comment 2 of the reviewer 1, is to address the possible origin of renal lesions from nestin expressing progenitors in TSC. The revised manuscript provides the appropriate clarification, as the description of the results included in figure 7 was moved to the separate subsection.

Reviewer 1 comments on responses to Reviewer 2 comments:

The reviewer commented for the editor only and was satisfied with the authors responses.

Reviewer #3 (Remarks to the Author):

Cho et al. have been responsive to the initial set of criticisms and the paper is strengthened as a result. A number of issues remain.

1. Without genetic evidence, e.g., using CRISPR mutagenesis of the endogenous RHEB promoter, it

remains unproven that Notch regulates RHEB via NRE2 and NRE3 (though the data presented are consistent with the possibility that these elements may regulate RHEB, at least to some degree).

As mentioned in our first rebuttal letter, we agree that mutating endogenous NRE2 and NRE3 in angiomyolipoma cells would be ideal to prove the role of Notch1 in regulating expression of Rheb via NRE2/3. As indicated in the first rebuttal letter, we attempted to remove NRE2/3 sites from the promoter of endogenous Rheb in angiomyolipoma cells using CRISPR approach. However, despite our multiple efforts, described in the previous letter, we failed to achieve this goal. We showed data demonstrating that despite evident Cas9 expression in angiomyolipoma cells, this Cas9 did not cut DNA.

To understand the nature of this problem, we revised published articles on CRISPR-Cas9 system, and found evidence explaining our failure to successfully modify Rheb promoter. We provide this information for the reviewer's and editor's consideration below.

Malina et al., (*PAM multiplicity marks genomic target sites as inhibitory to CRISPR-Cas9 editing. Nat Commun. 2015 Dec 8;6:10124*) reported that the presence of multiple protospacer adjacent motifs (PAM) interferes with efficient Cas9 cleavage while editing DNA.

We found that each NRE2 and NRE3 region of Rheb promoter contains eleven PAM sites each, spanning the region of 37 base pairs (shown below; bold black or bold red indicates NRE2 or NRE3 sequences; red indicates individual PAM sites outside of NRE2/3 sequences; bold red indicates PAM sites within NRE2 or NRE3 sequences).

NRE2

ccaatcaccgcac**ctcccac**ctattgccccgc**cct**gc
ccaatcaccgcac**ctcccac**ctattgccccg**ccc**tgc
ccaatcaccgcac**ctcccac**ctattgcc**ccg**ccctgc
ccaatcaccgcac**ctcccac**ctattgc**ccc**gccctgc
ccaatcaccgcac**ctcccac**ctattg**ccc**cgccctgc
ccaatcaccgcac**ctcccac**ctattgccccgc**ccct**gc
ccaatcaccgcac**ctcccac**ctattgccccgc**ccct**gc
ccaatcaccgcac**ctcccac**ctattgccccgc**ccct**gc
ccaatcaccgcac**ctcccac**ctattgccccgc**ccct**gc
ccaatca**ccg**cac**ctcccac**ctattgccccgc**ccct**gc
ccaatcaccgcac**ctcccac**ctattgccccgc**ccct**gc

NRE3

ccgctcctgtttcccca**atgggag**atggagttt**ccg**
ccgctcctgtttcccca**atgggag**at**ggg**agtttccg

ccgcctcctgtttccccca**atgggag**atggagtttccg
 ccgcctcctgtttccccca**atgggag**atggagtttccg
 ccgcctcctgtttccc**cca**atgggagatggagtttccg
 ccgcctcctgtttcc**CCC**a**atgggag**atggagtttccg
 ccgcctcctgtttc**CCC**ca**atgggag**atggagtttccg
 ccgcctcctgttt**CCC**cca**atgggag**atggagtttccg
 ccgcct**cct**tgtttccccca**atgggag**atggagtttccg
 ccg**cct**cctgtttccccca**atgggag**atggagtttccg
ccgcctcctgtttccccca**atgggag**atggagtttccg

The multiple PAM sites within NRE2/3 regions appear to correlate with poor Cas9 activity and ultimately lead to lack of efficient DNA cleavage at these two regions (as discussed/shown in our first rebuttal letter).

In addition, four continues guanines (GGGG) leads to poor CRISPR activity, poor yield for oligo synthesis and the tendency to form guanine tetrad, a secondary structure, making the guide sequence less accessible for target sequence recognition (*Wong et al., WU-CRISPR: characteristics of functional guide RNAs for the CRISPR/Cas9 system. Genome Biology 2015, 16:218*). Because two out of seven Rheb NRE2 guides designed using broad CRISPR designer tool at <http://portals.broadinstitute.org/gpp/public/> contained four continues guanines (NRE2 G46-AS 5'AAAC agggc**gggg**caataggtgggC-3'; NRE2 G50-AS 5'AAAC agggc**gggg**caataggtgggC-3') at 5' end in guide sequence domain (gRNA), we believe that the presence of these guanines also correlated with failure of CRISPR approach to target the Rheb promoter.

Furthermore, we used the website (<http://crispr.wustl.edu/>), shown to be useful in identifying functional gRNA candidates, to determine whether NRE2 and NRE3 regions of the Rheb promoter contains any sequences that could be used to design such gRNAs (*Wong et al., 2015, Genome Biology*). This database contains the experimentally validated gRNAs of 18,635 human and 20,354 mouse genes, and the design tool, which improves gRNA efficiency and efficacy through identifying the functionally active gRNAs with the appropriate structural features (http://crispr.wustl.edu/cgi-bin/gRNA_predict/custom_gRNA.cgi). WU-CRISPR returns gRNAs that have a potency score higher than 50, i.e. 50% likelihood of efficiency. The potency score indicates likelihood of the gRNA potency on a scale from 0 to 100. WU-CRISPR tool did not identify any efficient gRNA for NRE2 region and returned only one antisense gRNA sequence for NRE3 region with the potency score of 54 (5'-gaaactcatctcccattgg-3') that starts at base 18 of 39 nucleotide long sequence (ccgcctcctgtttcccc**caatgggag**atggagtttccg; bold red indicates gRNA start/bold black indicates NRE3 region).

In summary, the difficulties in targeting NRE2 and NRE3 regions of Rheb promoter through CRISPR approach appear to be a result of many overlapping intrinsic issues related to specific characteristics of the NRE2 and NRE3, making editing these sites through CRISPR extremely challenging, if not impossible.

We would like to emphasize that we provided evidence of Notch-dependent regulation of Rheb by using alternative Notch shRNA approach and overexpression of activated form of Notch1. We tested activity of Rheb luciferase promoter and directly demonstrated Notch-dependent regulation of the activity of Rheb promoter. We found that two NRE sites, NRE2 and NRE3, are equally involved in regulation of Rheb promoter (**Fig. 2D-i**). The basal activity of NRE2 and NRE3 mutants was reduced by 2-fold (**Fig.2D-i**). We demonstrated also that the activity of double NRE2/3 mutant is similar to single NRE2 or NRE3 mutants (**new Figure 2D-i**). **These new results suggest that both sites are equally important for Notch1-depend regulation of Rheb and that altering one of these sites is sufficient to abolish this regulation.** As mentioned before, NRE2 and NRE3 are responsible for approximately fifty percent of basal activity of Rheb promoter, indicating that other transcription factors are involved in regulation of Rheb promoter.

We demonstrated that overexpression of N1ICD and MAML1 increases the expression of endogenous Rheb (**Fig.2D-ii**). We verified that lowering expression of Notch1 reduced expression of endogenous Rheb, Notch1 and Hes1 (**Fig. 2D-iii-iv**). The reduced expression of Rheb at the protein level was associated with the reduced levels of Rheb mRNA. Finally, we have data indicating binding of Notch1 to the NRE2 and NRE3 sites on Rheb promoter in non-synchronized and synchronized cells (**Fig. 2C-ii-iv, 2E-i-iii and 4B**). Together these data support the role of Notch1 in regulating or tuning expression of Rheb.

Parentetically, it would be of interest to know if double mutation of NRE2 and NRE3 results in reduction of RHEB promoter activity below the ~50% reduction seen with single mutations in these sites.

We generated NRE2/NRE3 double mutant and determined that activity of this mutant is similar to single NRE2 or NRE3 mutants, suggesting that two sites are required for Notch1-dependent transactivation of Rheb, and that altering either one is sufficient to eradicate this regulation. These new data are included in Figure 2D-i.

Figure 2D-i

2. There is additional confusion caused by other nomenclature used by the authors. In Figure 2Dii and 2Dii, it appears that the label N1ICD refers to the transfected Myc-tagged activated Notch1 protein, whereas in Figure 2Diii N1ICD presumably refers to endogenous Notch polypeptides in each cell type that are recognized by an antibody against an epitope located in the intracellular region of Notch1. Most of these polypeptides do not correspond to activated Notch1 (they are mostly the transmembrane subunit of furin-cleaved full-length Notch1) and some other label should be used. By contrast, the V1744 antibody used in Figure 4D and 4G is specific for N1ICD, and if this name is to be applied to polypeptides in any of the Western blots presented, it should be applied here. It also is confusing to call any sequence that resembles an RBPJ binding site an NRE; this name should be reserved for sites that are proven to be functionally important.

To address the reviewer's concern, we included additional western blotting data on the level of cleaved Notch1 at V1744 (Figure 2D-iii) and changed former "N1ICD" description to "Notch1" wherever Notch1 was detected using antibody recognizing C-terminus of Notch 1 receptor (Figure 2D-iii, 4D and 4G). We also changed the language in our manuscript and substituted "NRE" with "Potential RBPJ Binding Site" or "RBPJ Binding Site". Because our data indicates that NRE2 and NRE3 are Notch1 responsive sites, we

Figure 2D-iii

kept their description as in the original manuscript.

3. In Figure 2D, it appears that the effects of shRNA knockdown of Notch1 on Rheb is quite a bit more dramatic at the protein level (Figure 2Diii) than it is at the RNA level (Figure 2Div). The description of the data doesn't acknowledge this discrepancy, which implies that the interaction between Notch and Rheb protein levels may involve mechanisms beyond transcription regulation of Rheb by Notch. In a similar vein, in Figure 4G, it is evident that N1ICD (activated Notch1) is sharply downregulated on day 1, yet there appears to be no change in Rheb protein levels at this time point. How long does it take for N1ICD levels to fall? Is Rheb a stable protein (and therefore the fall in Rheb lags well behind the drop in N1ICD)?

We agree with the reviewer that interaction between Notch1 and Rheb may involve regulation of Rheb at the protein level. *Makovski et al. (Farnesylthiosalicylic acid (salirasib) inhibits Rheb in TSC2-null ELT3 cells: A potential treatment for lymphangiomyomatosis, International Journal of Cancer, 130, 1420-1429, 2012)* determined, using *Tsc2*-null rat uterine leiomyoma cells (ELT3), that Rheb protein is stable for at least 26 hours and undergoes initial degradation at 31 hour. Consistent with this report our data (not shown) support stability of Rheb protein with its half-life of 3-4 hours. Therefore, data in the figures 2B and 6C, showing significant reduction of Rheb at the protein level within 90 minutes, support the notion that Notch1 also regulates Rheb protein degradation.

Because we found this observation very interesting, we are initiating a separate study to determine how Notch1 regulates Rheb protein degradation and whether it involves similar mechanisms shown for degradation of β -catenin protein in stem and progenitor cells (*Kwon et al., 2011, Notch post-translationally regulates β -catenin protein in stem and progenitor cells. Nature Cell Biology 13, 1244–1251 (2011) doi:10.1038/ncb2313*). In this study, Notch titrated negatively β -catenin via Numb-dependent lysosomal trafficking of the Notch- β -catenin complexes for degradation. Furthermore, Notch-dependent regulation of β -catenin degradation did not require cleavage of Notch1 and formation of very unstable N1ICD. Due to extensive nature of this regulation the results of this study will be published in the separate manuscript. However, we discussed the possible regulation of Rheb protein by Notch1 in the discussion section of the current revision of our manuscript and referred to the discovery by *Kwon et al., 2011*.

Other minor comments

In Figure 4I, the microarray data are redundant with 4J and can be deleted.

We removed microarray data from panel 4I.

Figure S3 is mislabeled.

We correct this error.

The source of the cleaved Notch1 antibody is not given in the methods

The source of cleaved Notch1 antibody is added to the methods.

The subheadings in the results' section were modified to meet editorial requirements.

REVIEWERS' COMMENTS:

Reviewer #1 (Remarks to the Author):

Thank you for your responses.
I believe that the revised paper is now a good shape.

Reviewer #3 (Remarks to the Author):

The authors have been responsive to the second set of critiques. The precise nature of Rheb regulation by Notch remains uncertain, but crosstalk between Notch and Rheb is convincing. Some modification in the main conclusions drawn, particularly in the abstract, may be indicated, as certain aspects of the proposed model remain to be proven.

We thank the reviewers for positive comments. Please find our responses to reviewers' comments below in blue.

Reviewer #3 (Remarks to the Author):

The authors have been responsive to the second set of critiques. The precise nature of Rheb regulation by Notch remains uncertain, but crosstalk between Notch and Rheb is convincing. Some modification in the main conclusions drawn, particularly in the abstract, may be indicated, as certain aspects of the proposed model remain to be proven.

We incorporated into revised manuscript a sentence indicating that certain aspects of our proposed model remain to be confirmed. This statement is now included in the discussion section.

“In summary, we have provided direct and indirect evidence for the significant role of the Rheb-Notch-Rheb loop in maintaining multipotent properties of Tsc-null cells and TSC tumorigenesis, although certain aspects of the proposed model remain to be confirmed.”